# IBI-CCS: a regional high-resolution model to simulate sea level in Western Europe

Alisée A. Chaigneau[1,2], Guillaume Reffray[2], Aurore Voldoire[1], Angélique Melet[2]

[1]CNRM UMR 3589, Météo-France/CNRS, Toulouse, France
[2]Mercator Ocean International, Toulouse, France

*Correspondence to*: Alisée A. Chaigneau (achaigneau@mercator-ocean.fr)

**Abstract.** Projections of coastal sea level (SL) changes are of great interest for coastal risk assessment and decision making. SL projections are typically produced using global climate models (GCMs), which cannot fully resolve SL changes at the coast due to their coarse resolution and lack of representation of some relevant processes (tides, atmospheric surface pressure forcing, waves). To overcome these limitations and refine projections at regional scales, GCMs can be dynamically downscaled through the implementation of a high-resolution regional climate model (RCM). In this study, we developed the IBI-CCS (Iberian-Biscay-Ireland Climate Change Scenarios) regional ocean model based on a 1/12 ° northeastern Atlantic NEMO ocean model configuration to dynamically downscale CNRM-CM6-1-HR, a GCM with a ¼ ° resolution ocean model component participating in the Coupled Model Intercomparison Project 6th Phase (CMIP6) by the Centre National de Recherches Météorologiques (CNRM). For a more complete representation of the processes driving coastal SL changes, tides and atmospheric surface pressure forcing are explicitly resolved in IBI-CCS in addition to the ocean general circulation. To limit the propagation of climate drifts and biases from the GCM into the regional simulations, several corrections are applied to the GCM fields used to force the RCM. The regional simulations are performed over the 1950 to 2100 period for two climate change scenarios (SSP1-2.6 and SSP5-8.5). To validate the dynamical downscaling method, the RCM and GCM simulations are compared to reanalyses and observations over the 1993-2014 period for a selection of ocean variables including SL. Results indicate that large-scale performances of IBI-CCS are better than those of the GCM thanks to the corrections applied to the RCM. Extreme SLs are also satisfactorily represented in the IBI-CCS historical simulation. Comparison of the RCM and GCM 21st century projections shows a limited impact of increased resolution (1/4° to 1/12°) on SL changes. Overall, bias corrections have a moderate impact on projected coastal SL changes, except in the Mediterranean Sea, where GCM biases were substantial.

## 1 Introduction

Sea level (SL) changes are a major threat for coastal and low-lying regions. Higher SLs can lead to coastal flooding, erosion, salinization of surface waters and groundwater, degradation of coastal ecosystems such as mangroves and coral reefs, and permanent submergence of land and human settlements (Oppenheimer et al., 2019). Risks associated with sea level rise (SLR) are even more important because coastal regions are subject to increasing anthropogenic pressure, with 10 % of the world's population living in low elevation coastal zones (McGranahan et al., 2007). In Europe, the coastal population represents 50 million people (Neumann et al., 2015). Without adaptation measures, the annual number of European people exposed to coastal flooding could reach 1.5 to 3.6 million by the end of the century and the associated expected annual damage could reach 90 to 960 billion euros (Vousdoukas et al., 2018a). Projections of coastal SL changes are thus of great interest for coastal risk assessment and decision making.

Variations of the SL at the coast result from the superposition of global mean sea level (GMSL), regional SL and local SL changes (e.g. Fox-Kemper et al., 2021; Oppenheimer et al., 2019; Woodworth et al., 2019). GMSL rise is driven by the thermal expansion of the ocean and the transfer of water mass from the cryosphere and land to the ocean (Church et al., 2013; Slangen

et al., 2017). At regional scales, spatial variations of SL changes are mainly due to changes in dynamic sea level (DSL), i.e. changes in ocean circulations and the associated ocean heat, salt and mass redistribution within the ocean (Forget and Ponte, 2015; Meyssignac et al., 2017a). At coastal scales, variations of SL are mostly related to tides, waves and atmospheric surges (e.g. Melet et al., 2018; Woodworth et al., 2019). Atmospheric surges are defined here as SL changes due to surface atmospheric pressure and local SLR caused by the wind, known as wind setup. At the coast, deviations from the GMSL can therefore be substantial (Kopp et al., 2014; Meyssignac et al., 2017b; Melet et al., 2020). Local relative SL change information is thus required by policy makers.

SL projections are typically based on global climate models (GCMs) (Oppenheimer et al., 2019; Church et al., 2013; Slangen et al., 2014). However, the typical CMIP5/6 (Coupled Model Intercomparison Project 5/6th Phase) model resolutions (mostly 1 °) do not allow fine scale processes to be resolved. These coarse resolutions limit the realism of the representation of coastal dynamical processes influencing SL changes at the coast (Woodworth et al., 2019), potentially leading to substantial biases. For example, van Westen et al., 2020 demonstrated for the Caribbean Sea that adequate regional projections of SL changes can only be obtained with ocean models that capture mesoscale processes. In addition, GCMs do not explicitly resolve key processes driving SL changes at the coast (e.g. waves, tides).

Dynamical downscaling methods can be used to refine GCM projections at regional scales by increasing the spatial resolution of the model and by explicitly including more processes (e.g. tides, atmospheric surface pressure forcing). Such methods rely on the implementation of a high-resolution regional climate model (RCM) driven by GCM outputs. Several studies have investigated future changes in ocean temperature, salinity, circulation and SL using dynamical downscaling in various regions (e.g. Mathis et al., 2013; Adloff et al., 2018, 2015; Shin and Alexander, 2020; Gomis et al., 2016; Macias et al., 2018). Some have focused specifically on SL projections (Hermans et al., 2020b; Liu et al., 2016; Zhang et al., 2017; Jin et al., 2021). Hermans et al., 2020b show the influence of dynamical downscaling for DSL projections over the twenty-first century on the northwestern European shelf using two GCMs. For the scenario with the highest radiative forcing by the end of the century (RCP8.5), they found that the downscaled DSL changes can be up to 15.5 cm smaller than in the GCM simulations. These differences are found in some coastal areas owing to unresolved processes in the GCM. For the North Pacific, Liu et al., 2016 performed a dynamical downscaling with 3 different CMIP5 GCMs. They showed that the downscaled SL changes can differ up to 10 cm from the GCM changes in coastal areas. Zhang et al., 2017 demonstrated the benefits of dynamical downscaling for Australian SL projections with a better representation of ocean gyre circulation and currents. Jin et al., 2021 used the dynamical downscaling method with 8 different GCMs to provide a modeling protocol to produce climate projections at low computational cost. Their results reveal greater spatial details in the downscaled simulations with differences up to 15 cm compared to the GCM simulations. Most of these studies have used low-resolution GCMs (around 1 ° spatial resolution), which explains the large differences in the projected sea level changes between GCMs and RCMs. However, few studies have employed higher resolution GCMs (e.g. ¼ ° over the ocean ≈ 28 km at the equator) in a dynamical downscaling framework. Hermans et al., 2020b have downscaled two GCMs with different ocean grid resolution. They concluded that the impact of dynamical downscaling is expected to be larger if the GCM has a lower resolution (typical CMIP5/6 model resolution of 1 °). The use of a GCM with a higher spatial resolution is nevertheless interesting for both the ocean (¼ °, eddy-permitting) and atmosphere. In particular, the atmospheric forcing applied to the regional ocean model is of higher resolution, which increases the realism of the forcing. For instance, the intensity of the atmospheric low-pressure systems and the spatial patterns generating extreme SL episodes should be better reproduced.

GCMs exhibit various biases when compared to observations (e.g. Flato et al., 2013). Because GCMs are used to force RCMs, these biases could propagate into regional simulations and be an important source of regional biases and uncertainties for the

projections (Takayabu et al., 2016; Dosio, 2016). To overcome this problem, bias corrections can be applied to the GCM outputs before using them as forcing when performing dynamical downscaling (e.g. Shin and Alexander, 2020). A simple method for bias correction is to shift the GCM data by its mean bias from a reference period. This method is used with a seasonal bias correction on the sea surface temperature in Adloff et al., 2015 and seasonal bias correction on the sea surface height (SSH) in Adloff et al., 2018, but has never been applied to 3D variables in the case of a dynamic downscaling method. Delta correction or anomaly forcing is another commonly used method: the GCM projected changes are added to a reference past state from reanalysis data or a climatology (Jin et al., 2021; Adloff et al., 2015). Other methods exist, such as rescaling the data with a factor (Macias et al., 2018 on the winds) or individually adjusting different ranges of a distribution. Emergent constraint methods also exist to overcome model biases and reduce the uncertainties of the projections (Chen et al., 2020; Grinsted and Christensen, 2021; Forster et al., 2021).

The aim of this study is to present a regional ocean model that will be used for analyzing the sensitivity of SL changes, particularly extreme SL changes, to methodological choices and representation of processes. The methodology employed is a regional dynamical downscaling of simulations from a 1/4° resolution GCM participating in the CMIP6. Bias corrections are applied to the GCM outputs (2D and 3D variables) before using them as forcings when performing the dynamical downscaling. The high-resolution regional ocean model (1/12°) includes coastal processes such as tides and surface atmospheric pressure forcing in addition to the ocean general circulation (DSL). Thanks to these included processes and the high frequency outputs of the RCM, it will be possible to use the regional simulations in a future study to investigate the projections of extreme SLs over the same region. In the current study, the configuration is presented along with its evaluation and the added value of the regional vs global model. From this perspective, we assess the influence on modeled regional SL changes of: (1) the dynamical downscaling i.e. the increased model resolution and a more complete representation of coastal processes driving SL changes (tides, sea level pressure forcing) and (2) bias corrections of GCM-forcing fields. For this purpose, the regional simulations are compared to original GCM simulations over the historical period and the 21st century using two climate change scenarios. The methodology presented in this paper could subsequently be applied to produce an ensemble of simulations using different CMIP6 global models as parent models to provide projections of sea level changes and related uncertainties.

This paper is organized as follows: the dynamical downscaling setup, correction methods, simulations performed and description of SL in the simulations are presented in Sect. 2. The dynamical downscaling method is evaluated in Sect. 3.1. Sect. 3.2 shows the RCM and GCM projections over the 21st century for two extreme greenhouse gas concentration scenarios of CNRM-CM6-1-HR: SSP5-8.5 and SSP1-2.6 (O'Neill et al., 2016). In Sect. 3, the added value of the dynamical downscaling and the impact of the applied bias corrections are assessed for the simulation of past and future ocean conditions with a focus on SL. The discussion and conclusions of the study are presented in Sect. 4.

## 2 Methods

The regional ocean model (RCM) IBI-CCS (Iberian-Biscay-Ireland Climate Change Scenarios) is developed and presented in Sect. 2.1.2. In this study, IBI-CCS is forced by the CNRM-CM6-1-HR CMIP6 climate model (Sect. 2.1.1) using ocean and atmospheric outputs at the lateral and air-sea boundaries of the regional IBI domain (Sect. 2.2). Several corrections are applied to the GCM forcings to limit the propagation of climate drifts and biases into the regional simulations (Sect. 2.2.1). CNRM-CM6-1-HR was chosen mostly for its high resolution over the ocean (¼ °, eddy-permitting) compared to the typical ocean grid resolution of CMIP6 models (1°). Other reasons for the choice of CNRM-CM6-1-HR are its high resolution over the atmosphere (½ °) and its high frequency outputs. These two aspects are very important for the modeling of extreme SLs, which will be the purpose of a future study.

## 2.1 Model and configuration

The ocean component of the global climate model CNRM-CM6-1-HR and the ocean regional model IBI-CCS are based on the 3.6 version of NEMO and rely on the Boussinesq approximation and a hydrostatic equilibrium (Madec et al., 2017). Only the horizontal ocean resolution is increased in IBI-CCS compared to CNRM-CM6-1-HR. Both vertical grids contain 75 z-levels with a resolution decreasing from about 1 meter in the upper 10 meters to more than 400 meters in the deep ocean. A partial step representation (Barnier et al., 2006) is implemented for the bottom ocean cell to better represent the bathymetry and the model benefits from variable volume-free sea surface.

### 2.1.1 Global climate model, CNRM-CM6-1-HR

The global climate model used to force the regional ocean model is the CNRM-CM6-1-HR ocean-atmosphere coupled model, developed jointly by the Centre National de Recherches Météorologiques (CNRM) and the Centre Européen de Recherche et de Formation Avancée en Calcul Scientifique (CERFACS). CNRM-CM6-1-HR contributes to CMIP6. The ocean component grid of this GCM has a ¼ ° horizontal nominal resolution (≈ 12-25 km at 25-65° N) with refinements in the equatorial band. CNRM-CM6-1-HR is a high-resolution model compared to the typical ocean grid resolution of CMIP6 models of 1 ° (≈ 50-100 km at 25-65° N). CNRM-CM6-1-HR is the high-resolution version of the 1 ° resolution CNRM-CM6-1, which is described in Voldoire et al., 2019; Roehrig et al., 2020. Some comparisons of CNRM-CM6-1-HR and CNRM-CM6-1 are included in Sect. 3 to assess the impact of the increased resolution between the two GCMs. Seawater thermodynamics uses a polynomial approximation of TEOS-10 (Roquet et al., 2015), therefore the prognostic variables are the absolute salinity and conservative temperature. As the RCM does not use the same approximation for the equation of state (section 2.1.2), the GCM outputs are converted to in-situ temperature and practical salinity to be used in the RCM. The vertical mixing of tracers and momentum uses the turbulent kinetic energy scheme (Gaspar et al., 1990; Blanke and Delecluse, 1993) and the internal wave-induced mixing parameterization of de Lavergne et al., 2020. The advection of the tracers is computed with the centered second order formulation combined with the limiter of Zalesak, 1979. The solar penetration is parameterized according to a four-band scheme. The NEMO ocean model and a sea-ice scheme GELATO are coupled to a land-atmosphere model using the OASIS-MCT coupler (Craig et al., 2017). The atmospheric component is the global atmospheric model ARPEGE-Climat 6.3 with a horizontal resolution of ½ ° (≈ 24-50 km at 25-65° N) at the Equator (Roehrig et al., 2020).

Four GCM simulations are used to prepare the forcings for the RCM: a historical run (1850-2014) forced by observed greenhouse gas concentrations, a preindustrial control run forced by fixed preindustrial conditions representative of the 1850s over 300 years, and scenarios (2015-2100) based on alternative trajectories for future emissions. In this paper, due to the computational cost of the simulations, we focus on only two greenhouse gas concentration scenarios. To obtain the most contrasting results possible, the two extreme scenarios of CNRM-CM6-1-HR included in Tier 1 of ScenarioMIP were chosen: SSP5-8.5 and SSP1-2.6 with, respectively, a very high and low radiative forcing by the end of the century (O'Neill et al., 2016). The SSP5-8.5 scenario relies on a fossil-fuel based world development leading to an Earth radiative imbalance of 8.5 W m$^{-2}$ in 2100. This scenario corresponds approximately to the CMIP5 RCP8.5 scenario. The SSP1-2.6 scenario has an approximately 50 % chance of following the Paris agreement of a limited warming below 2 °C by the end of the century and corresponds to the CMIP5 RCP2.6 scenario (Lee et al., 2021).

**2.1.2 Regional ocean model IBI-CCS at 1/12° resolution**

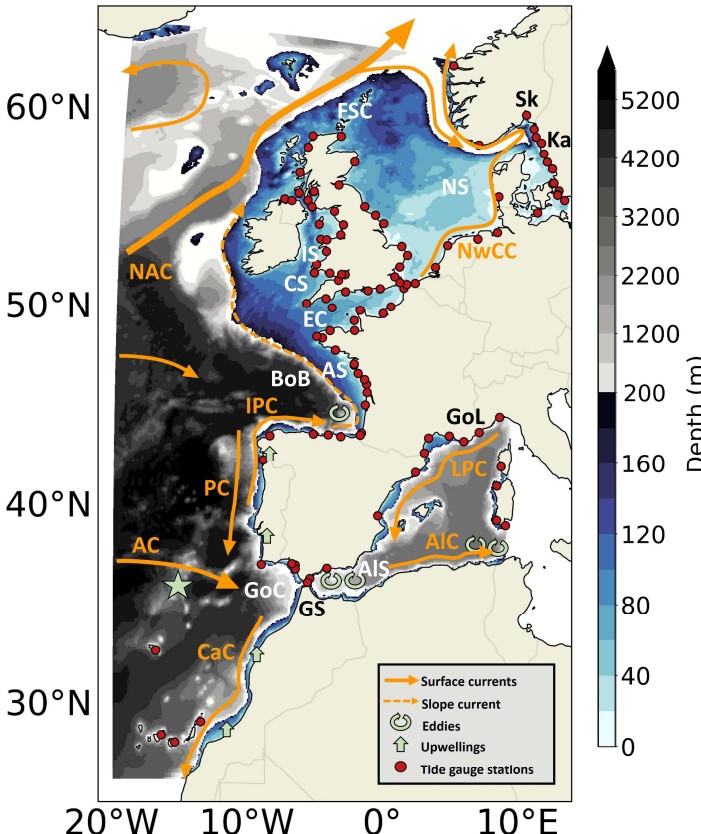

**Figure 1: Bathymetry (m) and schematic description of main oceanographic features in the IBI domain. The main surface dynamical features shown are: the North Atlantic Current (NAC), the Azores Current (AC), the Canary Current (CaC), the Portugal Current (PC), the Iberian Poleward Current (IPC), the Norwegian Coastal Current (NwCC), the Liguro Provençal Current (LPC), the**
**Algerian Current (AlC). Some geographical features are mentioned: the Bay of Biscay (BoB), the Armorican Shelf (AS), the English Channel (EC), the Irish Sea (IS), the Celtic Sea (CS), the North Sea (NS), the Faroe - Shetland Channel (FSC), the Kattegat (Ka), the Skagerrak (Sk), the Gulf of Cadiz (GoC), the Alboran Sea (AlS), the Gulf of Lion (GoL) and the Gibraltar Strait (GS). The star indicates the zone where a TS diagram is performed in Sect. 3.1.3. The red dots represent the tide gauge stations of the zone used in Sect. 3.1.6. Note that the color scale for the bathymetry is not linear. The shelf break (defined by the 200 m isobath) is indicated by**
**the change in the colormap).**

The configuration used for the regional IBI-CCS simulations is based on a curvilinear grid at a 1/12 ° horizontal resolution. The domain covered by IBI-CCS is the Iberian-Biscay-Ireland (IBI) zone, extending from 25° N to 65° N (model resolution ≈ 4-8.5 km) and 21° W to 14° E. This region includes the northeastern Atlantic Ocean, the North Sea and the western Mediterranean Sea (Fig. 1). The IBI zone is covered in the framework of the Copernicus Marine Service (CMEMS), with a
1/36 ° real-time system and a 1/12 ° reanalysis IBIRYS, including assimilation of observations and bias corrections. The configuration of IBIRYS (described in Baladrón et al., 2020 and validated in Levier et al., 2020) was used to perform the IBI-CCS simulations. The reanalysis has shown very good performances against observations in Levier et al. 2020 for several variables used to validate the IBI-CCS simulations in Sect. 3.1: temperature, salinity, currents, sea level. As IBIRYS and IBI-CSS share the same modeling framework, better results are not expected with IBI-CCS_corr than with IBIRYS. Therefore,
IBIRYS is considered as the reference for the IBI zone in the study.

A variety of physical oceanographic processes are found in the IBI region (Sotillo et al., 2015; Maraldi et al., 2013). First, the zone contains strong variations of bathymetry, with a wide continental shelf in the northern part of the domain (North Sea, English Channel) and a tight continental shelf in the southern part (Spain, Portugal, Morocco, Mediterranean Sea) (Fig. 1). The northwestern, deeper part of the IBI region is mainly driven by the North Atlantic Current (NAC). Along the continental
slope, a poleward slope current flows from the Portuguese coasts to the north of Ireland with slope oceanic eddies along the northern Iberian coast. On the continental shelves, large energetic tides are found, particularly in the English Channel, Celtic

and Irish Seas. In the southern part of the domain, two main physical features are found: strong summer upwellings along the Portuguese and Moroccan coasts and Gibraltar Strait. In Gibraltar Strait, exchanges between the Atlantic Ocean and Mediterranean Sea occur and drive mesoscale eddies in the Alboran Sea (Fig. 1).

The main improvement of this configuration in comparison to the GCM (Sect. 2.1.1) is the inclusion of processes driving SL changes in the coastal ocean, such as tides and atmospheric pressure forcing in addition to the ocean general circulation (DSL). Tides are included in the model by calculating the astronomical tidal potential and the tidal harmonic forcing. SSH and barotropic velocities tidal components are added through the open boundaries as the sum of 11 components provided by FES2004 (Lyard et al., 2006) and TPXO7.1(Egbert and Erofeeva, 2002): diurnal components (K1, O1, P1 and Q1), semi-
diurnal constituents (M2, S2, N2 and K2), long-period tides (Mf and Mm) and a nonlinear component M4. Tides were validated in the same way in the 1/12° configuration with the IBIRYS reanalysis as in section 3.1 of Maraldi et al., 2013 (1/36° configuration). The validation of M2 tidal amplitude is also provided in the Supplementary Materials (Fig. S3). In addition to these high frequency processes added in the RCM, some physical parameterizations also differ from those of the GCM. Seawater thermodynamics uses a polynomial approximation of EOS-80 (Fofonoff and Millard Jr, 1983). Therefore, the RCM
requires in-situ temperature and practical salinity from the GCM. Vertical mixing is parameterized according to a k-ε model implemented in the generic form proposed by (Umlauf and Burchard, 2003; Reffray et al., 2015). Although tides are treated explicitly, the mixing induced by internal tides is not completely resolved in the RCM and the parameterization of de Lavergne et al., 2020 is also activated in the RCM (as in the GCM). The advection of tracers is computed with the QUICKEST scheme developed by Leonard, 1979 combined with the limiter of Zalesak, 1979. The solar penetration parameterization is based on a
five-band exponential scheme. Finally, air-sea turbulent fluxes are calculated in the model using ECMWF-IFS bulk formulations (ECMWF, 2014; Brodeau et al., 2017).

**2.2 Regionally downscaled simulations**

The regionally downscaled simulations are performed over the 1950-2100 period. 1950-1970 is considered as the spin-up period, after which surface and intermediate waters have reached a quasi-equilibrium (not shown here). The historical regional
simulation therefore starts in 1970 and ends in 2014. Climate change simulations using scenarios described in Sect. 2.1.1 are run from 2015 to 2100.

The near surface atmospheric state variables from the GCM used to force the regional ocean model are the three-hourly 2m-air temperature (t2m), 2m-specific humidity (q2m), 10m-wind, short and long wave radiations, precipitation and six-hourly
atmospheric pressure at SL. The open boundary conditions (OBCs) are prescribed at the lateral boundaries of the regional domain each month using the GCM 3D ocean temperature, salinity, currents and 2D sea surface height (SSH). Temperature, salinity and baroclinic velocities of the GCM are prescribed at the frontier point of the domain and a buffer zone of 10 grid points relaxes the internal regional solution to the prescribed boundary values. For the SSH forcing, the GCM SSH is not prescribed directly, but rather enters into a Flather-type algorithm (Flather and Davies, 1976). The RCM is initialized using
the GCM state of January 1950 for temperature, salinity, currents and SSH. The monthly runoff outputs taken from the ocean component of the GCM are also prescribed to the RCM.

Global climate models are typically subject to drift (i.e long-term change independent of internal variability or external forcings, especially in the deep ocean, Gupta et al., 2013) and to substantial biases (Flato et al., 2013, Fasullo, 2020). To
prevent the regional simulations from inheriting the drift and biases of the GCM, several corrections were applied to the GCM outputs before prescribing them to the RCM. Two different simulations are performed to assess the impact of these corrections (Table 1). The first is the raw simulation referred to as IBI-CCS_raw, in which atmospheric and OBC forcings, initial

conditions and runoffs are taken directly from the GCM outputs without any correction by extraction and interpolation on the 1/12° regional grid. The second is referred to as IBI-CCS_corr, with corrections of the GCM forcings (Sect. 2.2.1). A third
simulation called IBI-ERAi (Sect. 2.2.2) is performed as a reference simulation to validate the IBI-CCS_raw and IBI-CCS_corr in Sect. 3.

| | Atmospheric forcings | | Open boundary conditions forcings | | Initial conditions | Runoff forcings | |
|---|---|---|---|---|---|---|---|
| **Simulation** | Fields | Frequency | Fields | Frequency | Forcing | Fields | Frequency |
| **IBI-CCS_raw (1950-2100)** | CNRM-CM6-1-HR | 3hours | CNRM-CM6-1-HR | 1 month | CNRM-CM6-1-HR | CNRM-CM6-1-HR seen by the ocean component | 1 month |
| **IBI-CCS_corr (1950-2100)** | CNRM-CM6-1-HR drift (t2m, q2m) and bias (t2m, q2m, radiative fluxes) corrected | 3hours | CNRM-CM6-1-HR drift and bias corrected (T,S,SSH) + SSH setting in Mediterranean Sea | 1 month | CNRM-CM6-1-HR drift and bias corrected (T,S,SSH) | TRIP (river routing model of CNRM-CM6-1-HR) | 1 day |
| **IBI-ERAi (1993-2104)** | ERAinterim | 3hours/ 1 day | GLORYS2V4 | 1 day | GLORYS2V4 | daily observations, simulated data and climatology (Baladrón et al., 2020) | 1 day |

**Table 1: Regional IBI-CCS and IBI-ERAi simulations forcings and settings and their corrections when applicable.**

### 2.2.1 IBI-CCS_corr simulation

**Drift correction**

Due to its high resolution and therefore high computational cost, CNRM-CM6-1-HR has a particularly short spin-up time of 250 years. As a consequence, the GCM is subject to larger drifts than coarser resolution CMIP6 class models with longer spin-up integrations. For example, the sea surface temperature drift can reach +1 °C in 2100 at 60 °N of the western boundary (not shown here). To avoid the GCM drift effect on regionally simulated long-term trends, the drift is removed from the GCM
outputs before using them to force the RCM. As shown in Irving et al., 2021, in most cases a linear fit of the preindustrial control simulation is sufficient to evaluate the drift in CMIP6 models. In Fox-Kemper et al., 2021 (Supplementary Materials Chapter 9, section 9.SM.4.2), the dynamic sea level "zos" and global mean thermosteric sea level "zostoga" variables have also been corrected from the drift linearly. For example, in Hermans et al., 2021, the drift for the variable "zostoga" appears nearly linear for most CMIP6 models. For the CNRM-CM6-1-HR variables concerned by drifts (Table 1), a linear fit is indeed
appropriate (not shown here). The CNRM-CM6-1-HR model drift is estimated at each grid point by a linear fit of the full time series of the pre-industrial control simulation. Then, the linear fit is subtracted from the corresponding historical simulation and projections at each time step and grid point. The drift is removed from the air temperature and specific humidity for atmospheric forcings, and from 3D temperature, 3D salinity and 2D SSH for ocean forcings (Table 1). The drift is also removed from the global mean thermosteric sea level (variable "zostoga") of section 2.3.2 using the same method.


**Bias correction**

To limit the GCM bias propagation into the IBI-CCS_corr projections, a simple seasonal mean bias correction (Xu et al., 2019; Adloff et al., 2015, 2018; Macias et al., 2018) is applied to the GCM outputs before using them to initialize and force the RCM. Bias adjustments allow a more realistic ocean mean state representation and conserve the GCM variability in the regional

simulations. This method relies on a stationarity hypothesis, i.e. biases do not depend on the mean state and are thus assumed to be the same in historical and scenario simulations (Krinner and Flanner, 2018).

The bias corrections applied to the IBI-CCS_corr forcings are based on the oceanic reanalysis GLORYS2V4, considered here as the reference dataset. The GLORYS2V4 reanalysis distributed by CMEMS has been largely validated in Garric et al., 2017. To apply these bias corrections, monthly mean differences between the GCM and GLORYS2V4 are computed over the 1993-2014 period. Then, the mean seasonal cycle of biases is subtracted from the GCM outputs at each time step and each grid point for the past, present and future periods. This method is applied to the ocean 3D temperature, 3D salinity and 2D SSH used at the OBCs and as initial conditions (Table 1). The velocity field quickly adjusts to these corrections. The amplitude of the temperature and salinity bias corrections for the western boundary of the domain is provided in the Supplementary Materials (Fig. S4). For the GCM atmospheric outputs, the surface (2 m) air temperature and specific humidity, as well as short and long wave radiation, are seasonally bias-corrected using the ERAinterim reanalysis (Berrisford et al., 2009) as a reference dataset using methodology similar to the one described for the OBCs. The ERAinterim reanalysis was chosen to maintain consistency with the corrections applied on the ocean, as ERAinterim was used to force the GLORYS2V4 reanalysis employed to bias-correct the ocean GCM outputs.

**Modification of the river forcing**

In CNRM-CM6-1-HR, the river discharges of the river routing component (TRIP, 0.5 °) are interpolated on the ocean model grid (0.5 °). Despite the global water budget is conserved, this interpolation results in large errors in regional runoff amounts (Voldoire, 2020). For instance, a large overestimate is found for the runoff of the Rhone, one of the major rivers in the IBI western Mediterranean domain, which causes a 4 psu freshwater bias in CNRM-CM6-1-HR in this region (Fig. S2). For the IBI-CCS_corr simulation, the river runoff forcing is thus taken directly from the daily runoff simulated by TRIP (Table 1) and interpolated on the 1/12 ° grid. Therefore, in IBI-CCS_corr, the RCM does not receive the same amount of runoff as in IBI-CCS_raw and the ocean component of CNRM-CM6-1-HR.

**Sea Surface Height tuning in the Mediterranean Sea**

In the Mediterranean Sea, the excess of evaporation over precipitation and river runoff is compensated by a net inflow of fresh Atlantic waters. These waters are transformed into denser waters and leave the Mediterranean Sea as deep currents through Gibraltar Strait. Therefore, realistic exchanges through the strait are of great importance for modeling volume transport and water mass properties. The net transport through Gibraltar Strait is directly related to the difference in pressure between the Atlantic Ocean and Mediterranean Sea, which is linked to the difference of SSH between the two basins (Soto-Navarro et al., 2010). The bias corrections (Sect. 2.2.1) allow the ocean mean state of GLORYS2V4, and thus a more realistic representation of the water masses than in the GCM, to be obtained. However, GLORYS2V4 has a mean SSH bias of approximately -0.1 m in the Mediterranean Sea in comparison to the Mean Dynamic Topography observations from CNES-CLS-18 (Mulet et al., 2021); see Supplementary Materials (Fig. S5). In consequence, the bias correction applied on the SSH (Sect. 2.2.1) is not sufficient to obtain a more accurate net transport through Gibraltar Strait. A tuning has been added to improve it in the IBI-CCS_corr simulation. A SSH corrective value of +0.1 m is thus applied to the east Mediterranean boundary at each time step and boundary grid point to compensate for the GLORYS2V4 bias and to obtain a proper difference of SSH between the two basins. The mass correction is added to the local T/S values.

**2.2.2 IBI-ERAi simulation**

In addition to the reference IBIRYS reanalysis provided by CMEMS (Sect. 2.1.2), another reference simulation called IBI-ERAi was performed based on the same configuration. IBI-ERAi is a free simulation (no data assimilation) forced

by GLORYS2V4 and ERAinterim, which are also the two reanalyses used to diagnose bias corrections in IBI-CCS_corr (Sect. 2.2.1). IBI-ERAi is therefore considered the best simulation to be directly compared to IBI-CCS_corr in order to evaluate the dynamical downscaling of the GCM.

## 2.3 Correction of the main SL components in the regional simulations

The regional simulations performed in this study are intended to be used to investigate the projections of extreme SLs in particular. The frequency of extreme SLs depends on total SL rise rather than just the ocean dynamic component (Menéndez and Woodworth, 2010; Vousdoukas et al., 2018b), hence additional SL change components need to be incorporated in the model.

### 2.3.1 Transfer of water mass from the cryosphere to the ocean

Barystatic SLR (Gregory et al., 2019) is dominated by the mass loss of glaciers and ice sheets (Oppenheimer et al., 2019). While GCM can, to some extent, represent surface mass balance processes, they cannot yet account for dynamic mass loss. In CNRM-CM6-1-HR, the glaciers and Greenland mass losses are underestimated compared to projection assessments (Table 2).

| Contributions to GMSL (in m) | scenario | CNRM-CM6-1-HR (2081-2100 relative to 1986-2005) | Oppenheimer et al., 2019 (2081-2100 relative to 1986-2005) | Hock et al., 2019 (2081-2100 relative to 2000) |
|---|---|---|---|---|
| Antarctica | SSP5-8.5 | 0.09 | 0.10 [0.02-0.23] | |
| Antarctica | SSP1-2.6 | 0.05 | 0.04 [0.01-0.10] | |
| Greenland | SSP5-8.5 | 0.06 | 0.12 [0.07-0.21] | |
| Greenland | SSP1-2.6 | 0.03 | 0.07 [0.04-0.10] | |
| Glaciers | SSP5-8.5 | 0.04 | 0.16 [0.09-0.23] | 0.14 |
| Glaciers | SSP1-2.6 | 0.03 | 0.10 [0.04-0.16] | 0.09 |
| Total | SSP5-8.5 | 0.19 | 0.38 | |
| Total | SSP1-2.6 | 0.11 | 0.21 | |

**Table 2: Projected global mean changes in SL mass dominating contributions for the SSP5-8.5 and SSP1-2.6 scenarios in CNRM-CM6-1-HR, Oppenheimer et al., 2019 and Hock et al., 2019.**

To estimate SLR over the IBI region due to glaciers and ice sheets mass loss, we used the fingerprints of these contributions to scale their global mean contribution to the regional domain. To that end, we used spatial fingerprints from Grinsted et al., 2015, expressed as a percentage of the GMSL contribution for the different land-ice components. The Antarctic ice sheet, Greenland ice sheet and glaciers' contributions to GMSL simulated in CNRM-CM6-1-HR and derived from Oppenheimer et al., 2019 and Hock et al., 2019 are given in Table 2. In CNRM-CM6-1-HR, the Antarctic contribution to GMSL is similar to that of Oppenheimer et al., 2019 and Hock et al., 2019 (although probably not for the right reason), whereas the contributions from the Greenland ice sheet and glaciers are clearly underestimated. The regional contributions obtained after applying the spatial fingerprint are presented in Fig. 2. The global mean contribution is weighted by a factor 120% for the Antarctic, 50% for the glaciers and is close to zero for Greenland (i.e. the effects of the Greenland ice mass loss on the IBI zone are considered null on average due to its distance from the ice sheet). As it turns out, the sum of all the regional land ice mass contributions estimated from the literature (grey solid line in Fig. 2) is very close to the CNRM-CM6-1-HR simulated contributions for both scenarios (grey dashed line in Fig. 2). Therefore, ultimately, no corrections concerning the mass change terms have been applied to the GCM.

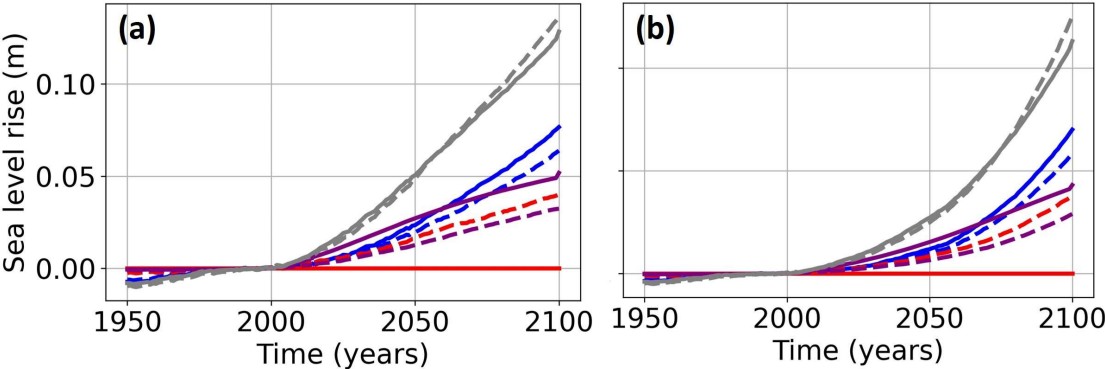

**Figure 2: Projected changes up to 2100 (relative to 1986-2005) of dominating barystatic SLR contributions scaled to the IBI domain using their spatial fingerprints for (a) SSP1-2.6 scenario (b) SSP5-8.5 scenario. The SL contributions evaluated here are: the Antarctic ice-sheet contribution (blue), the Greenland ice-sheet contribution (red), the glaciers contribution (purple), and their sum (grey). The dashed lines represent the initial CNRM-CM6-1-HR contributions and the solid lines the contributions based on Oppenheimer et al., 2019 for the Antarctic ice sheet and on Hock et al., 2019 for glaciers.**

### 2.3.2 Thermal expansion

In Boussinesq ocean models such as NEMO CNRM-CM6-1-HR or IBI-CCS, the ocean volume evolves according to the mass budget but does not change globally according to ocean density changes. The global mean thermosteric sea level (GMTSL) rise, which corresponds to a thermal expansion of the ocean, is therefore not explicitly represented in such models. As the GMTSL rise is a dominant contribution to the GMSL rise (Oppenheimer et al. 2019, Fox-Kemper et al. 2021), it has to be evaluated a posteriori from the simulated ocean density field (Greatbatch, 1994; Griffies and Greatbatch, 2012). As the water column cannot expand, the GMTSL rise cannot be prescribed directly to the RCM because any increase in volume can only result in an addition of mass. This addition of mass would be added directly at local temperature and salinity properties, which would increase the pressure gradient and result in an acceleration of the circulation close to the boundaries. The GCM GMTSL term stored in the variable "zostoga" is thus added a posteriori to the RCM modeled SL after having removed the drift (section 2.2.1).

### 2.3.3 Total SL in global and regional simulations

Although Boussinesq models do not represent the expansion of the water column, they are able to correctly reproduce the local steric effect (Griffies and Greatbatch, 2012) related to changes in the local density of the water column through the equation of state. As non-uniform density changes create pressure gradients, the ocean circulation is dynamically adjusted (e.g. thermal wind balance) and spatial gradients of DSL are simulated (Griffies and Greatbatch, 2012). Therefore, we conclude that the only missing SL term in the regional model comes from the GMTSL.

In the GCM, the total SL $\eta$ is diagnosed by:

$$\eta = \text{GMTSL (global process)} + \text{Sea Surface Height } = \text{ zostoga (GCM) } + \text{ SSH (GCM) } \quad (1)$$

where the Sea Surface Height (SSH) in the global model includes regional processes:

- the mass variations corresponding to the freshwater balance including:
  - the balance between evaporation and precipitation + river runoff
  - the transfer of water mass from the cryosphere and land to the ocean from the GCM (Sect. 2.3.1).
- the DSL, which corresponds to the variable 'zos'. The main drivers of DSL are steric SL (ocean density related) and manometric SL (mass related) components (Gregory et al., 2019). The steric SL itself can be decomposed into thermosteric and halosteric components related to changes in density due to temperature and salinity changes, respectively.

In the regional IBI-CCS_raw and IBI-CCS_corr simulations, the total SL $\eta$ is diagnosed by:

$$\eta = \text{GMTSL (global process)} + \text{Sea Surface Height} = \text{zostoga (GCM)} + \text{SSH (RCM)} \quad (2)$$

where the SSH in the regional model includes regional and coastal processes:

- mass variations
- the DSL, which corresponds to the variable "zos" of the GCM but which also includes tides, the barotropic effects due to SL pressure forcing.

Following Gregory et al., 2019, the DSL is corrected by the inverse barometer (IB) effect. The latter is computed based on the Stammer and Hüttemann, 2008 formulation. The IB effect is included in the presented DSL results unless stated otherwise. In the following sections the term "SL" refers to the total SL $\eta$.

## 3 Results

### 3.1 Historical simulations and validation of the ocean regional climate model

To validate the dynamical downscaling method, the IBI-CCS_raw, IBI-CCS_corr and CNRM-CM6-1-HR historical simulations are compared to the reanalysis IBIRYS, the IBI-ERAi (Sect. 2.2.2) regional simulation and observational datasets over the 1993-2014 period. The comparisons are performed at different time scales for a selection of ocean variables, including SL. Due to the chaotic nature of the climate system, GCMs do not follow the real-world internal variability chronology, but they should represent a climate internal variability statistically similar to the observed one. Consequently, only the model's ability to reproduce observed distributions is assessed. In this section, in addition to the validation of the IBI-CCS regional simulations, the added value of the dynamical downscaling (in terms of resolution and added physical processes) and of the bias corrections applied are investigated.

### 3.1.1 Thermosteric, Halosteric, Steric and Manometric SL

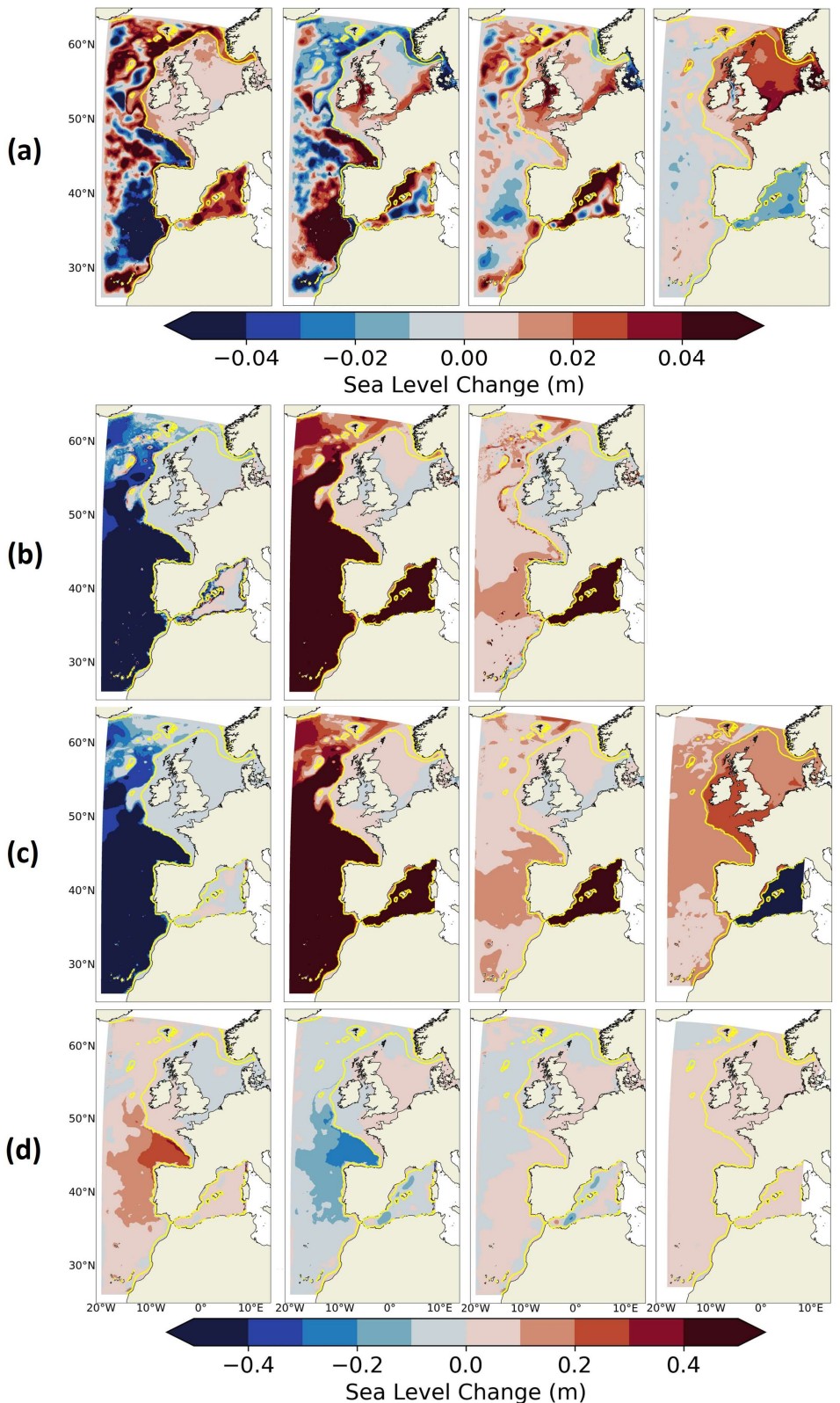

**Figure 3: Thermosteric (first column), halosteric (second column), steric (third column) and manometric (last column) SL bias over 1993-2014 between (a) IBI-ERAi and IBIRYS, to show biases in the IBI-ERAi simulation, and between (b) CNRM-CM6-1-HR and IBI-ERAi, (c) IBI-CCS_raw and IBI-ERAi, (d) IBI-CCS_corr and IBI-ERAi. Manometric SL biases between CNRM-CM6-1-HR and IBI-ERAi are not shown here as they mostly display the differences of bathymetry between the two models. The thermosteric, halosteric and steric components have been computed over 0-2000 m depth. Note the different colorbars in panel (a) and in panels (b), (c), (d). The shelf break (defined by the 200 m isobath is indicated in yellow).**

Figure 3 compares the main components of DSL (Sect. 2.3.3) averaged over the 1993-2014 period for the different simulations.

As the thermosteric and halosteric SL components are depth-integrated variables, the comparisons allow the heat and salt

content of the model to be validated between 0 and 2000 m, respectively. Differences between the reanalysis IBIRYS and IBI-ERAi highlight the biases of IBI-ERAi and do not exceed 10 cm, even in the deep ocean (Fig. 3a). Biases between CNRM-CM6-1-HR and IBI-ERAi are large (Fig. 3b). Indeed, in the Atlantic Ocean, the thermosteric SL is 40 cm too low due to a large cold bias. The GCM halosteric SL is 50 cm higher than its IBI-ERAi counterpart in the Mediterranean Sea because of a fresh bias due to the strong positive bias in the Rhone River discharge received by the GCM ocean component (Sect. 2.2.1). This fresh bias seems to spread in the Atlantic Ocean through Gibraltar Strait. In the Atlantic Ocean, the thermosteric and halosteric biases balance each other out, leading to small biases on the steric SL. However, this is not the case in the Mediterranean Sea, where the halosteric bias leads to steric biases of a larger amplitude. The large biases found in the GCM propagate into IBI-CCS_raw with the same amplitude (Fig. 3c). These biases are consistent with the cold sea surface temperature and fresh salinity biases provided in the Supplementary Materials (Fig. S1 and Fig. S2).

In IBI-CCS_corr, runoffs are taken directly from the river routing model to avoid the regional discrepancies present in the GCM and subsequently in IBI-CCS_raw simulations (Sect. 2.2.1). The change of runoff results in a considerable reduction of the halosteric bias in the Mediterranean Sea. The reduction of the biases on all the different SL components in IBI-CCS_corr (Fig. 3d) is consistent with the bias correction method used to correct the GCM forcings. Indeed, the 1993-2014 period was used to compute the biases between the GCM and the ocean and atmospheric reanalyses used to force IBI-ERAi. The applied corrections have therefore been well integrated into the model, as results for the 1993-2014 period are close to those of IBI-ERAi, especially for the steric and manometric SL components. Some thermosteric and halosteric biases still exist in IBI-CCS_corr (Fig. 3d, first and second column) in the Bay of Biscay (Fig. 1). These biases are related to the Mediterranean water outflow, which does not occur at exactly the same depth or with the same characteristics in IBI-CCS_corr and IBI-ERAi, as shown in the TS diagram in Sect. 3.1.3 (Fig. 5).

### 3.1.2 Circulation

**Surface Circulation**

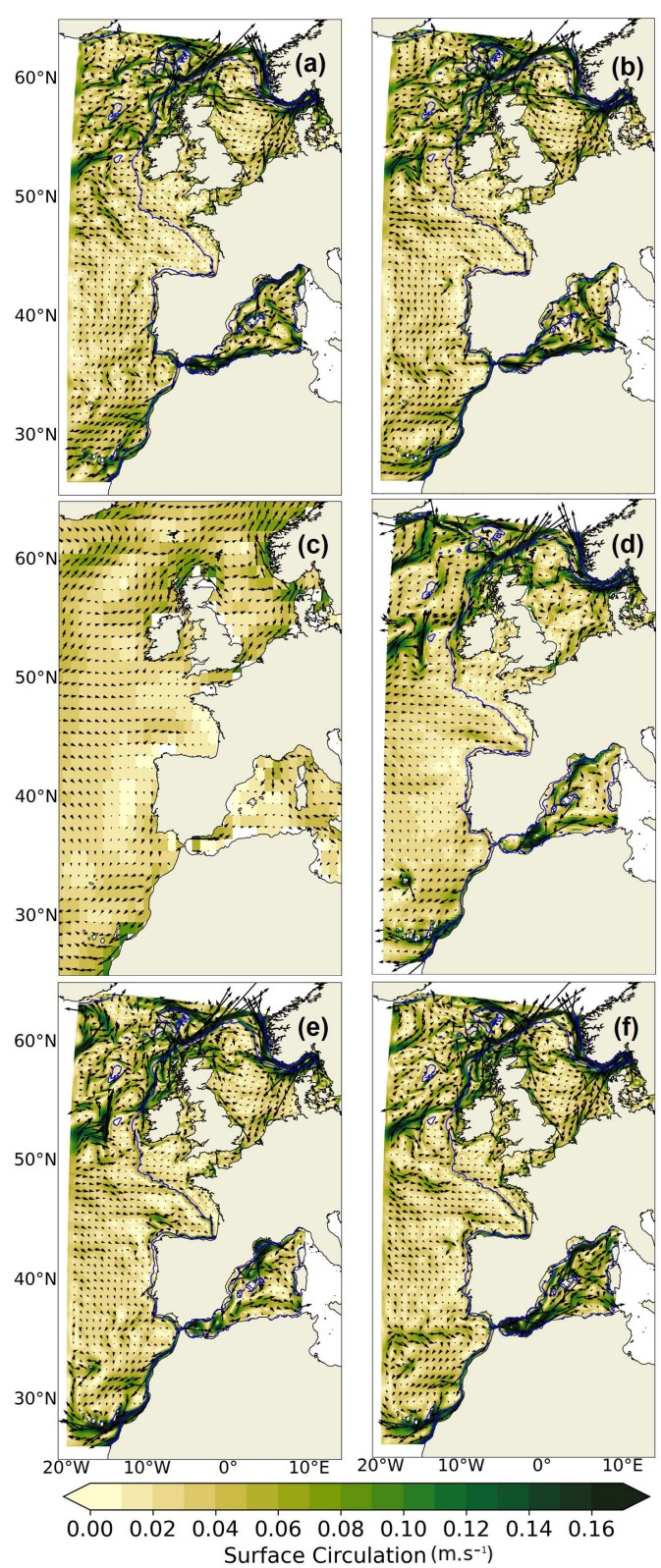

**Figure 4: Surface circulation (in m.s⁻¹) over the 1993-2014 period in (a) IBIRYS (b) IBI-ERAi (c) CNRM-CM6-1 (d) CNRM-CM6-1-HR (e) IBI-CCS_raw and (f) IBI-CCS_corr simulations. The shelf break (defined by the 200 m isobath is indicated in blue).**

The general ocean surface circulation averaged over 1993-2014 is illustrated in Fig. 4. The main regional surface dynamical features described in Maraldi et al., 2013; Sotillo et al., 2015 and captured in IBIRYS (Fig. 4a) and IBI-ERAi (Fig. 4b) are

420 also represented in all GCM and RCM simulations. In the Atlantic Ocean, the North Atlantic Current (Fig. 1) enters the IBI zone at 52°N on the western boundary and separates into several branches, with the main one flowing eastward north of the United Kingdom. The Norwegian Costal Current (Fig. 1) in the North Sea and along the Norwegian coasts, the Canary Current (Fig. 1) along the Moroccan coasts are other main currents correctly simulated by the GCMs and RCMs. The GCM CNRM-

CM6-1-HR and regional simulations show the Liguro Provençal Current (Fig. 1) flowing westward along the northern continental shelf in the Mediterranean Sea. However, owing to the large biases of temperature and salinity found in both GCMs in the Mediterranean Sea (Sect. 3.1.1 and Supplementary Materials Fig. S1 and Fig. S2), the surface circulation cannot be considered realistic in the basin in the corresponding simulations (Fig. 4c,d).

When comparing Fig. 4c and d, the impact of the increased resolution between the two GCMs is clear. Thanks to its higher ocean model resolution, CNRM-CM6-1-HR shows a more realistic regional circulation than the 1 ° GCM CNRM-CM6-1. The impact of the dynamical downscaling would therefore be significantly higher using a 1 ° typical CMIP resolution ocean model to force the RCM. However, the dynamically downscaled simulations (Fig. 4e,f) add even more spatial information compared to the GCM CNRM-CM6-1-HR (Fig. 4d). One of the major improvements in the IBI-CCS simulations is the emergence of an additional North Atlantic Current branch at 48° N, south of the major North Atlantic Current branch, as in the reference simulations (Fig. 4a,b). Another added value of the dynamical downscaling is seen with the poleward slope currents from the Iberian coasts up to Ireland, which do not exist in the GCM simulations. Along the Iberian coasts, where the southward Portugal Current and the northward Iberian Poleward Current (Fig. 1) co-exit (Cordeiro et al., 2018), the GCMs show no clear feature. Conversely, both the IBI-CCS_raw and IBI-CCS_corr simulations exhibit the Portugal Current and Iberian Poleward Current currents. These two currents are also found in IBI-ERAi with approximately the same amplitude, but not in the reanalysis IBIRYS for the Iberian Poleward Current. Finally, the gyre in the Alboran Sea (Fig. 1), just east of Gibraltar Strait, is represented in the regionally downscaled simulations but not in the GCMs.

IBI-CCS_corr and IBI-CCS_raw are now compared to assess the impact of bias corrections on the surface circulation (Fig. 4e,f). The major difference is the appearance of the eastern branch of the Azores Current (Fig. 1) in IBI-CCS_corr, at 35°N with a southward recirculation, as in the reference IBI-ERAi simulation. Another difference between IBI-CCS_corr and IBI-CCS_raw is the strengthening of the North Atlantic Current branch at 48° N in the corrected simulation, leading to a current closer to IBIRYS and IBI-ERAi. In the Mediterranean Sea, large differences between the IBI-CCS_corr and IBI-CCS_raw simulations are also found, with a strengthening of the circulation in IBI-CCS_corr.

**Transport through Gibraltar Strait**

| Model/Simulation | Period | Inflow transport | Outflow transport | Net transport |
|---|---|---|---|---|
| IBI-ERAi | 1993-2014 | 1.06 Sv | -0.46 Sv | +0.60 Sv |
| IBIRYS | 1993-2014 | 1.13 Sv | -0.50 Sv | +0.63 Sv |
| CNRM-CM6-1-HR | 1993-2014 | 0.54 Sv | -0.55 Sv | -0.04 Sv |
| IBI-CCS_raw | 1993-2014 | 0.40 Sv | -0.10 Sv | +0.30 Sv |
| IBI-CCS_corr | 1993-2014 | 0.76 Sv | -0.70 Sv | + 0.06 Sv |
| Soto-Navarro et al., 2010 | 2004-2009 | 0.81 Sv | -0.78 Sv | +0.04 Sv |
| Soto-Navarro et al., 2015 | 2004-2007 | | | +0.05 Sv |
| Adloff et al., 2015 | 1961-1990 | 0.85 Sv | -0.80 Sv | +0.05 Sv |

**Table 3: Transport through Gibraltar Strait in the different simulations in comparison to previous studies. Transports are positive eastward.**

As explained in Sect. 2.2.1, realistic exchanges through Gibraltar Strait have a strong influence on water mass properties and thus on SL over the northeastern Atlantic region. The values of net transport are presented for CNRM-CM6-1-HR and regional simulations in Table 3 and compared to estimates by Soto-Navarro et al., 2010, 2015; Adloff et al., 2015. Results must be

interpreted with caution as the computation of the fluxes was performed offline (Soto-Navarro et al., 2020). In IBIRYS, the

460 inflow transport (from Atlantic Ocean to Mediterranean Sea) through Gibraltar Strait is overestimated, while the outflow transport (from Mediterranean Sea to Atlantic Ocean) is underestimated. The resulting net transport is largely overestimated (Table 3 and Levier et al., 2020). In the GCM and IBI-CCS_raw simulations, both inflow and outflow transports are too weak. This is consistent with the surface circulation (Fig. 4), where the entering current at Gibraltar Strait in the GCM and IBI-CCS_raw simulations is weaker than in the reanalysis. The values of the net transport in the IBI-CCS_raw and GCM

simulations are different and are not comparable to the estimates by Soto-Navarro et al., 2010, 2015; Adloff et al., 2015. On the contrary, in IBI-CCS_corr, thanks to the SSH tuning applied at the Mediterranean boundary (Sect. 2.2.1), the net transport, inflow and outflow transports are close to the estimates by Soto-Navarro et al., 2010, 2015; Adloff et al., 2015 with a value of +0.06 Sv for the net transport (Table 3).

### 3.1.3 Water masses properties

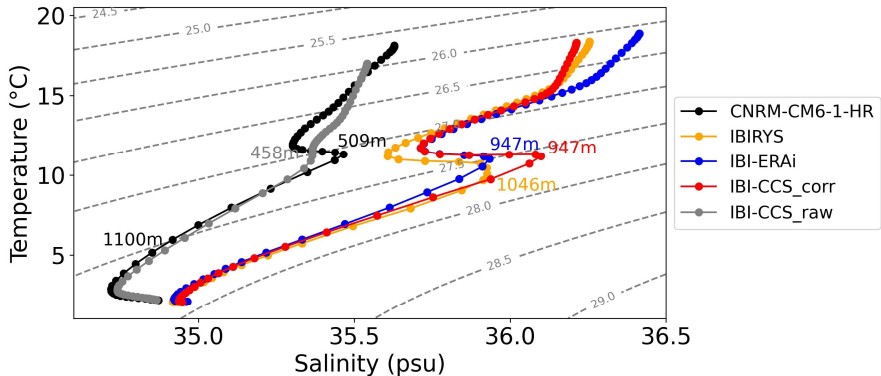

**Figure 5: TS diagram performed at the star location of Fig. 1 over the 1993-2014 period for the CNRM-CM-1-HR, IBIRYS, IBI-ERAi, IBI-CCS_corr, IBI-CCS_raw simulations.**

The impact of bias corrections for the representation of the Mediterranean water mass properties in the Atlantic Ocean is now assessed. Figure 5 compares the water mass thermohaline properties in the Atlantic Ocean west of Gibraltar Strait for the

475 GCM, IBI-CCS_raw and IBI-CCS_corr simulations to the IBIRYS and IBI-ERAi simulations. The location of the TS diagram (star location in Fig. 1) has been chosen far from the western frontier of the domain and in an area where the Mediterranean Outflow Water (MOW) has spread at a depth of around 1100 m (Bozec et al., 2011). Due to the large surface biases in temperature and salinity found in the GCM and IBI-CCS_raw in the Mediterranean Sea (Sect. 3.1.1 and Supplementary Materials Fig. S1 and Fig. S2), the water mass properties at Gibraltar Strait, and hence of the MOW, cannot be properly

modeled. Indeed, large biases are found in the Atlantic Ocean at the MOW depth: biases in temperature and salinity at 1100 m depth reach 4.5 °C and 1.5 psu respectively (Fig. 5). In contrast, in IBI-CCS_corr, where bias corrections are applied, the model is able to reproduce the transformation of fresh and warm surface Atlantic waters into dense and salty MOW, leading to a strong reduction of T/S biases (Fig. 5). In IBI-CCS_corr, MOW spreads westward at a depth of 950 m, in good agreement with IBI-ERAi and IBIRYS (Fig. 5). This indicates that bias corrections could lead to a change in the TS diagram shape and

water mass characteristics, particularly for the initially biased MOW and not only to a shift in temperature and salinity. Comparisons of 10-yr simulations with the two different river runoff forcings (Sect. 2.2.1) and with or without the SSH tuning in the Mediterranean Sea (Sect. 2.2.1) show that improvements in the T/S diagram are mostly due to the bias corrections (not shown). Thanks to the bias corrections, the water mass characteristics have been corrected, thus controlling the influence of the Mediterranean Sea on the Atlantic Ocean and preventing the propagation of the Mediterranean biases into the Atlantic

Ocean.

### 3.1.4 Mean Sea Surface Height

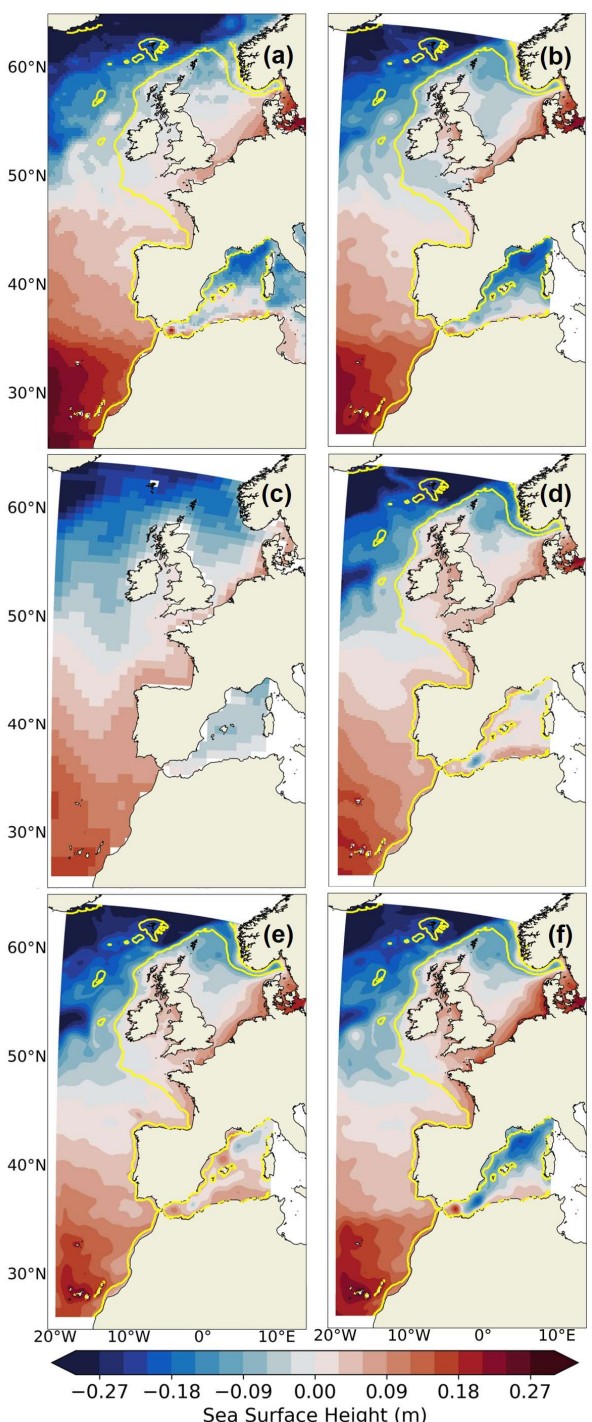

**Figure 6: Mean sea surface height over the 1993-2014 period in (a) MDT CNES-CLS-18 (b) IBI-ERAi (c) CNRM-CM6-1 (d) CNRM-CM6-1-HR (e) IBI-CCS_raw and (f) IBI-CCS_corr. Note that all panels here do not include the inverse barometer effect on sea level. The shelf break (defined by the 200 m isobath is indicated in yellow).**

The mean dynamic topography (MDT) gives the time mean sea surface height above the geoid due to ocean circulations. The dataset of reference for the MDT for the 1993-2012 period is the CNES CLS18 data set (Mulet et al., 2021). The CNES CLS18 MDT has a 1/8° resolution and is based on GOCE and GRACE data, altimetry and in-situ data, as well as on the GOCO05S geoid model. The observed MDT is comparable to the modeled time mean SL, referred to in this section as the mean sea surface height (MSSH).

Figure 6 compares the CNES CLS18 MDT to the MSSH for the different simulations, showing that the simulations reproduced the Atlantic main features of the observed MDT (Fig. 6a) well. Indeed, the Atlantic MSSH northwest to southeast gradient

associated with the North Atlantic Current (Fig. 1), subtropical and subpolar gyres is well reproduced in all the GCM and IBI-CCS simulations in comparison to observations and IBI-ERAi (Fig. 6). Along the coasts of the North Sea (Fig. 1) and eastern English Channel (Fig. 1), all GCM and IBI-CCS simulations show elevated MSSH similar to observations and the IBI-ERAi simulation. However, in the Bay of Biscay (Fig.1), in the GCM and IBI-CCS simulations, the MSSH is too elevated in comparison to observations and to the IBI-ERAi simulation (Fig. 6).

As for the surface circulation in Sect. 3.1.2, the impact of the increased resolution between the two GCMs (Fig. 6c,d) is clear, with more spatial information for the MSSH in CNRM-CM6-1-HR. Compared to CNRM-CM6-1-HR (Fig. 6d), the even higher resolution in IBI-CSS_raw (Fig. 6e) improves only slightly upon the MSSH in some coastal areas, such as the Celtic Sea (Fig. 1), Irish Sea (Fig. 1) and the western part of the English Channel.

The bias corrections applied in IBI-CCS_corr (Fig. 6f) improve the excessively low MSSH pattern at 53° N of the western boundary found in CNRM-CM6-1-HR and IBI-CCS_raw (Fig. 6d,e). The bias corrections also have a large impact in the Mediterranean Sea, where the GCM CNRM-CM6-1-HR and IBI-CCS_raw MSSH is overestimated. Indeed, in these simulations, the Atlantic waters flowing through Gibraltar Strait then flow northward toward the Balearic Islands and the Gulf of Lion, which is unrealistic according to observations and previous studies (Adloff et al., 2018). On the contrary, in IBI-CCS_corr, where bias corrections are applied, Atlantic waters are trapped in the Alboran gyre and then stick to the north African coast, as in the observed MDT and IBI-ERAi (Fig. 6). In the northern Mediterranean Sea, the low SL feature associated with the large gyre in the convection area of the Gulf of Lion is also well represented in IBI-CCS_corr in comparison to the MDT and IBI-ERAi.

### 3.1.5 SL interannual variability

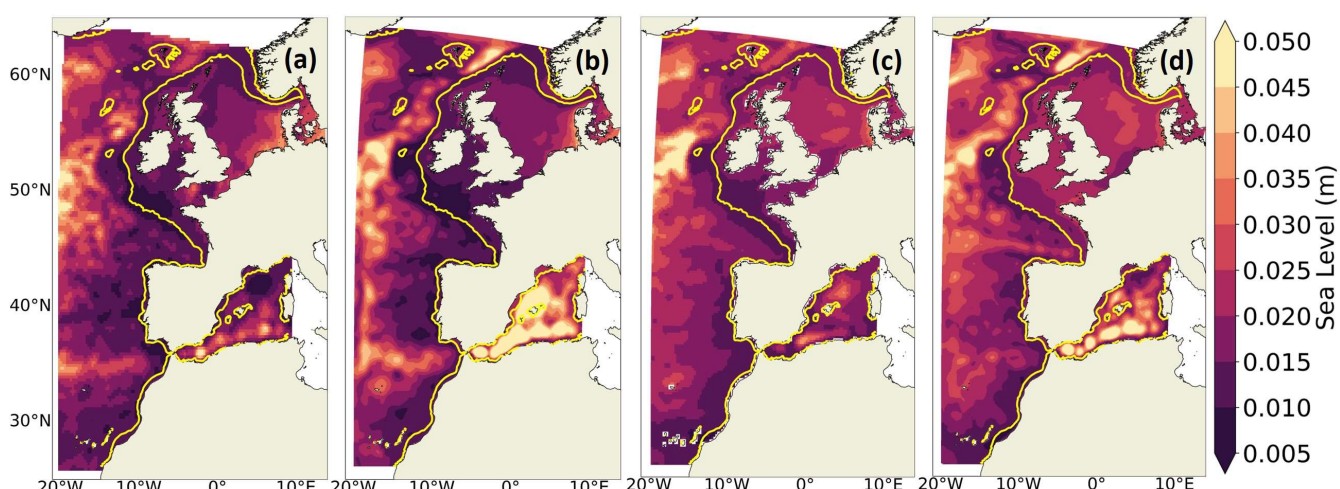

**Figure 7: SL interannual variability over the 1993-2014 period in (a) altimetry (b) IBI-ERAi (c) CNRM-CM6-1-HR and (d) IBI-CCS_corr. The interannual variability is computed as the standard deviation of the detrended annual mean SL. Note that all panels here do not include the inverse barometer effect on sea level. The shelf break (defined by the 200 m isobath is indicated in yellow).**

Here, the CNRM-CM6-1-HR and IBI-CCS_corr simulations are compared to the global gridded reprocessed SEALEVEL_GLO_PHY_L4_REP_OBSERVATIONS_008_047 altimetric observation product distributed by CMEMS. The product is provided at a 0.25 ° resolution starting from 1993 and is based on the combination of measurements from different altimeter missions (Taburet et al., 2021 and Pujet et al., 2020).

The large SL interannual variability associated with the zone's main currents, such as the North Atlantic Current, Azores Current, Algerian Current, and the Liguro Provençal Current of Fig. 1 is well represented in IBI-CCS_corr (Fig. 7d), as shown

in comparison with the altimetry product and IBI-ERAi (Fig. 7a, b). On the large continental shelf, the GCM and RCM both simulate a larger interannual variability than the altimetry product, except in the German Bight. Indeed, in the German Bight and north of the Netherlands, the altimetry product and IBI-ERAi show a relatively large interannual variability which is not present in CNRM-CM6-1-HR and IBI-CCS_corr. In both regional IBI-CCS_corr and IBI-ERAi simulations, the interannual variability in the Mediterranean Sea is very high, which is not the case in the GCM simulation and to a lesser extent in the altimetric product (Fig. 7).

### 3.1.6 Extreme SLs

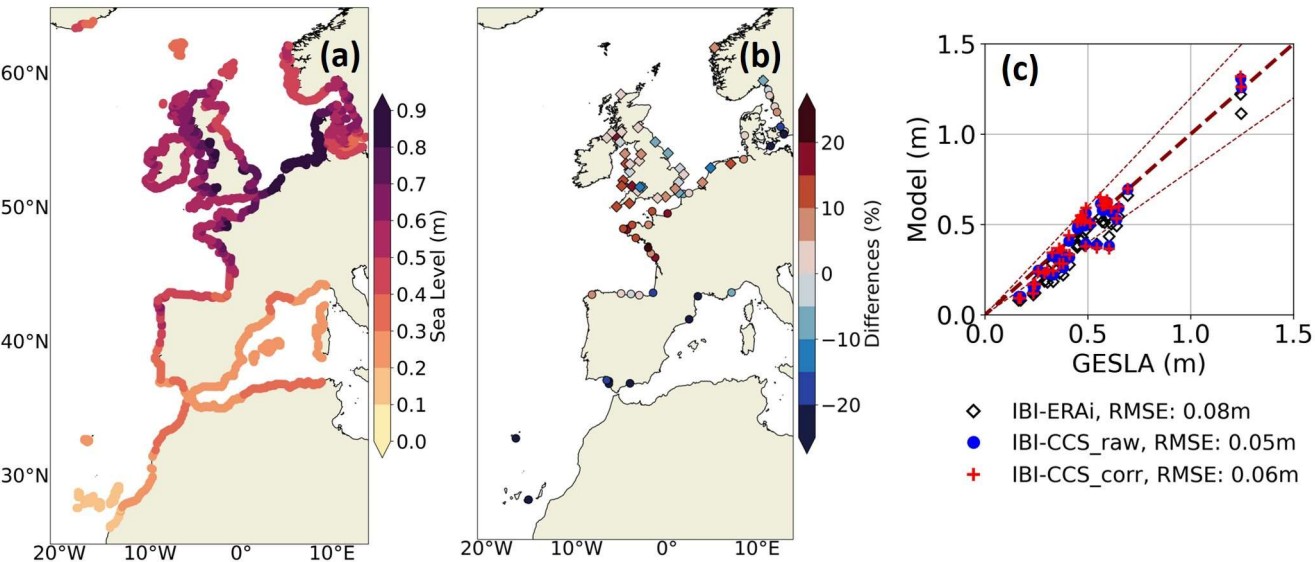

**Figure 8: (a) Non-tidal residuals 99th percentile values over the 1993-2014 period in IBI-CCS_corr. (b) Relative error of the non-tidal residuals 99th percentile of IBI-CCS_corr compared to GESLA TG data over the 1993-2014 period. Circles represent TG data at 1-hour frequency and the diamonds stand for TG data at higher frequency. (c) Scatter plot of simulated vs observed 99th percentile at TG stations in IBI-ERAi (white diamonds), IBI-CCS_raw (blue circles) and IBI-CCS_corr (red cross). The thin dark red dashed lines indicate the 20% error margin.**

For impact studies, it is even more crucial to obtain a good representation of SL extreme values. Extreme SLs of the IBI-CCS_raw and IBI-CCS_corr simulations are thus validated against tide gauge (TG) records and IBI-ERAi. The GESLA (Global Extreme Sea Level Analysis GESLAv2) dataset provides high-frequency (at least hourly) TG data records (Woodworth et al., 2017). The selected TG stations have a temporal data coverage of no less than 75% over the 1993-2014 period and are marked with red dots in Fig. 1.

Extreme SLs are investigated here with the 99th percentile based on hourly averaged outputs of the IBI-CCS model. The GCM CNRM-CM6-1-HR did not produce sufficiently high frequency outputs to assess such SL extreme events. In addition, CNRM-CM6-1-HR is not able to represent SL extremes properly, as they are highly related to tides, which are not represented in this model. Therefore, the explicit representation of processes such as tides is an important added value of the regional model. IBI-CCS and the IBIRYS reanalysis are based on the same configuration, including tide implementation, which has been validated in Levier et al., 2020 and Maraldi et al., 2013. Therefore, here the comparison focuses on the validation of non-tidal residuals (where tides are filtered from the SL time series). During extreme SL events, non-tidal residuals are dominated by atmospheric/storm surges.

Figure 8 shows that the non-tidal residuals 99th percentile in IBI-CCS_raw and IBI-CCS_corr is properly represented in comparison to TG data. Both IBI-CCS_raw and IBI-CCS_corr show performances similar to, but slightly better than, those of the reference simulation IBI-ERAi. Indeed, the error at the different TG stations rarely exceeds 20% (Fig. 8c) and the RMSE

do not exceed 5 cm. Errors found between IBI-CCS simulations and GESLA dataset are comparable to those of recent papers (Muis et al., 2020; Kirezci et al., 2020). The largest errors are found in the southern part of the domain, where the 99$^{th}$ percentile
of non-tidal residuals are the smallest. In conclusion, thanks to the atmospheric surface pressure forcing and high frequency outputs of the RCM, the regional simulations appear to be able to model extreme SLs correctly. The model will be further validated on extreme SLs; for example, in terms of return period, in a follow-up study.

### 3.2 Regional projections under climate change scenarios with a focus on SL

Here, the regional projections are presented for the SSP5-8.5 and SSP1-2.6 scenarios for the different variables validated in
the former section. Additionally, the effect of dynamical downscaling and of bias corrections on the projections is assessed, with a focus on SL changes.

### 3.2.1 Projected trend of regional mean total SL

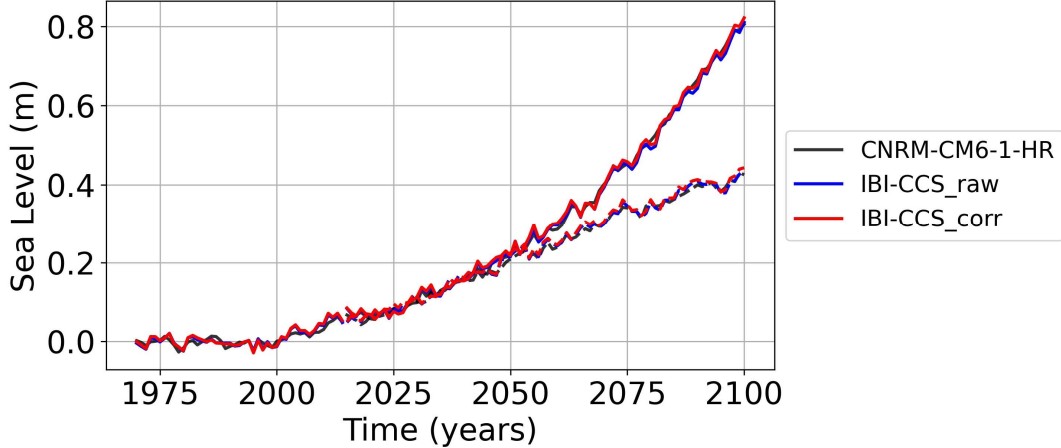

**Figure 9: Time series of annual mean SL changes (in m) averaged over the IBI domain for the historical period (1970-2014), SSP5-**
**8.5 (solid line) and SSP1-2.6 (dashed line) scenarios (referenced to 1986-2005) for the GCM and the two regionally downscaled IBI-CCS_raw and IBI-CCS_corr simulations.**

In Fig. 9, the historical and projected mean SL changes over the whole IBI domain are assessed for the SSP5-8.5 and SSP1-2.6 scenarios for the GCM and downscaled simulations. As explained in Sect. 2.3.3, none of the GCM and regionally downscaled simulations represent the spatial mean thermosteric effect on SL. It has thus been computed and added a posteriori.
By the end of the century, a mean SL increase of +80 cm is simulated over the IBI domain for the SSP5-8.5 scenario (relative to 1986-2005) and +40 cm for the SSP1-2.6 scenario. These values are close to the GMSL projections of +71 cm (RCP8.5) and +39 cm (RCP2.6) from Oppenheimer et al., 2019 over the same period.  In Fig. 9, the consistency of the trend of total regional mean SL between the global and regional simulations for the two scenarios validates the dynamical downscaling technique employed. It demonstrates that bias corrections do not impact the projected mean SL trend for both scenarios.

## 3.2.2 Projected changes in surface circulation


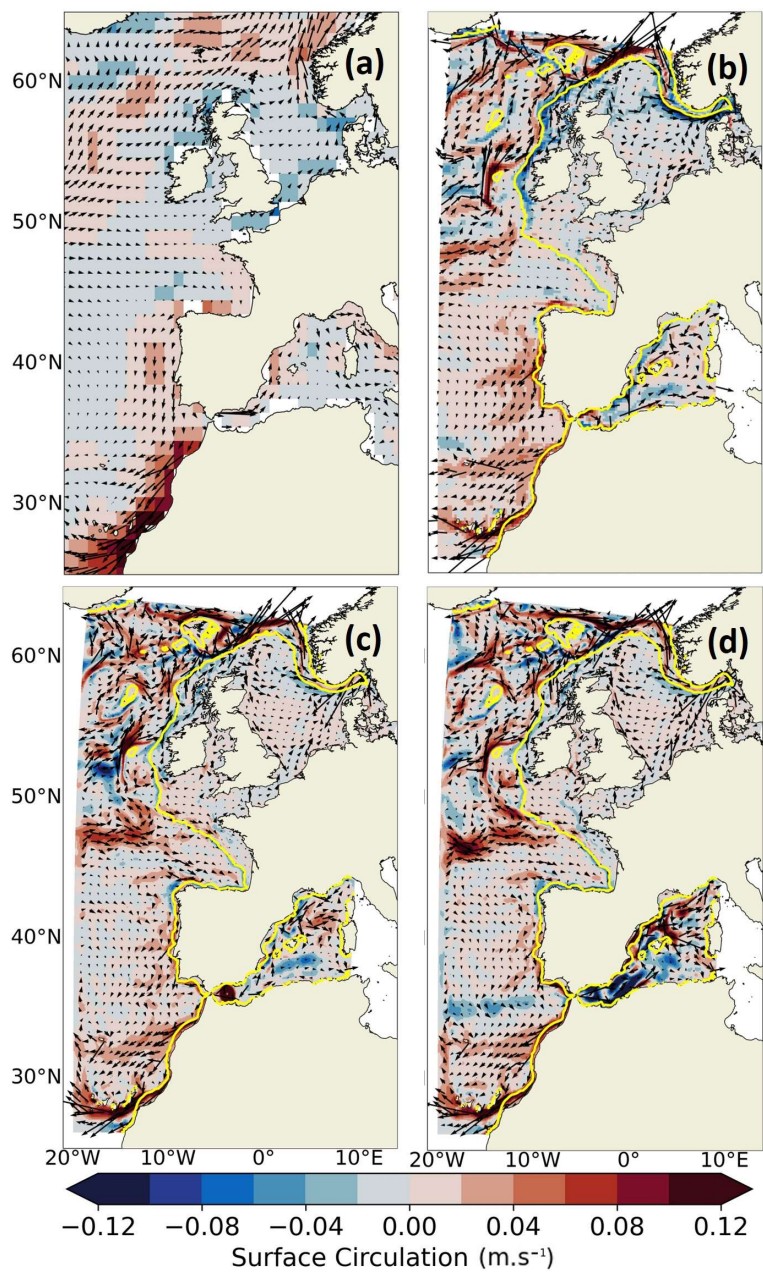

**Figure 10: Projected changes in surface ocean currents (in m.s⁻¹) for the 2081-2100 period (relative to 1986-2005) under the SSP5-8.5 scenario for (a) CNRM-CM6-1 (b) CNRM-CM6-1-HR (c) IBI-CCS_raw and (d) IBI-CCS_corr. The magnitude of surface currents changes is indicated by the color shading. The shelf break (defined by the 200 m isobath is indicated in yellow).**


Figure 10 shows projected changes in ocean surface currents under the SSP5-8.5 scenario in the two GCMs, IBI-CCS_raw and IBI-CCS_corr simulations. Projected changes generally agree well between the GCM and RCM simulations in terms of patterns, except in the Mediterranean Sea. While all the simulations show a strong intensification of the Portugal Current and Canary Current (Fig. 1), projected changes exhibit larger amplitudes in CNRM-CM6-1-HR (Fig. 10b) and IBI-CCS (Fig.
10c,d) than in CNRM-CM6-1 (Fig. 10a) thanks to their higher resolution. Indeed, in CNRM-CM6-1-HR and IBI-CCS, large changes are found in the north of the domain, with a strengthening of the branch of the North Atlantic Current (Fig. 1) around 48°N. The GCM also projects a strong decline of the Norwegian Coastal Current and in the North Atlantic Current branch flowing around the United Kingdom (Fig. 10b), which is not modeled in the RCM simulations (Fig. 10c,d). In both IBI-CCS simulations, the higher resolution adds more spatial information to the projections, such as the decline of the poleward current
from Iberia to Ireland (Fig. 10c,d). In the Mediterranean Sea, the four simulations show very different changes. These results

should be interpreted with caution, as the surface circulation of the historical simulations are not very realistic in the Mediterranean Sea. In the Mediterranean Sea, projected changes in the surface circulation are small in the GCMs and IBI-CCS_raw, whereas in IBI-CCS_corr, the projected changes are substantial and show a strong weakening of the Alboran gyre (Fig. 10d). In conclusion, both the resolution and bias corrections have a substantial impact on the projected changes in the surface circulation.

### 3.2.3 Projections of water mass properties

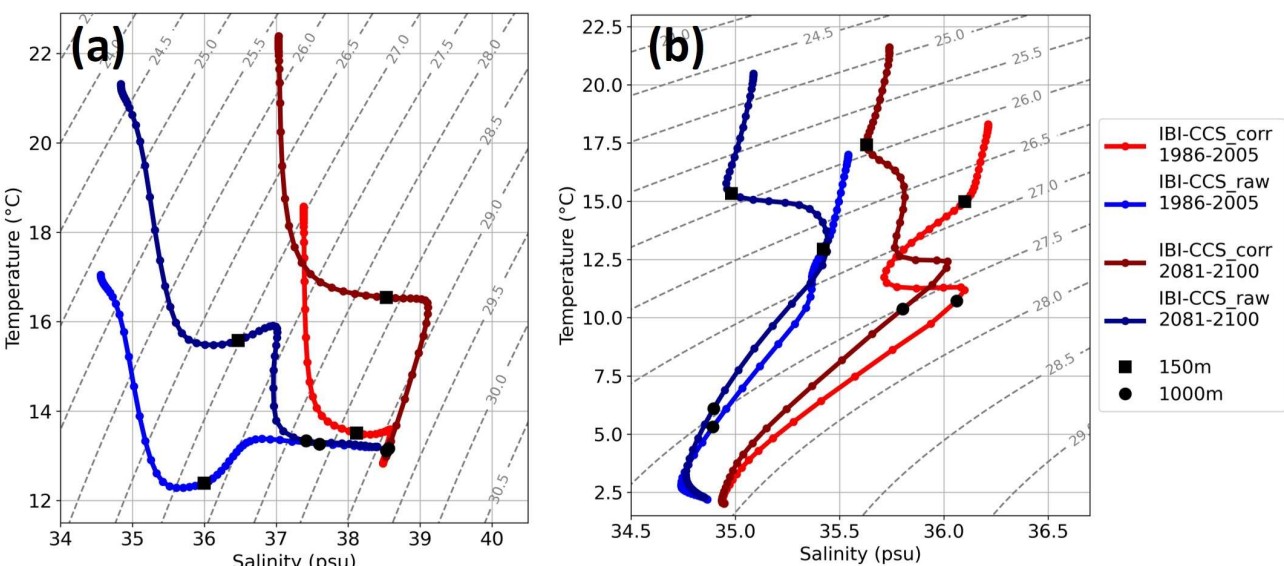

**Figure 11: TS diagram for the SSP5-8.5 scenario for the present (1986–2005) (light line) and future (2081–2100) (dark line) periods for IBI-CCS_corr (red) and IBI-CCS_raw (blue) (a) in the western Mediterranean Sea (b) at the star location of Fig. 1.**

The historical and projected changes of the water mass properties east and west of Gibraltar Strait are presented with a TS diagram in Fig. 11. Projections of the water masses TS characteristics in the western Mediterranean Sea (Fig. 11a) are in close agreement with those of Soto-Navarro et al., 2020 in IBI-CCS_corr, whereas projections from IBI-CCS_raw are totally out of the range of the CMIP5 models used in Soto-Navarro et al., 2020. Indeed, IBI-CCS_corr displays a strong warming of the upper 1000 m with a general decrease in density and also an abrupt change in the TS characteristics of intermediate and deep waters. In Soto-Navarro et al., 2020, the outlier CMIP5 simulations were excluded from the average T/S computation. The simulations accounted for are those with a good representation of the Mediterranean Sea. This is not the case of the GCM CNRM-CM6-1-HR, used here to force IBI-CCS_raw, which is probably an outlier simulation in this region. On the contrary, IBI-CCS_corr provides a much better representation of the Mediterranean Sea, which explains why the projected changes in IBI-CCS_corr are closer to the projections of Soto-Navarro et al., 2020 than those of IBI-CCS_raw. These results provide more confidence in the IBI-CCS_corr simulation.

The second TS diagram (Fig. 11b) is performed in the Atlantic Ocean, at the star location of Fig. 1. IBI-CCS_corr projections show a general warming and freshening of the water column. Mediterranean Outflow Water flowing westward in the Atlantic Ocean seem to be found at shallower depths at the end of the 21$^{st}$ century. When comparing the projections of IBI-CCS_raw and IBI-CCS_corr at a depth of 1000 m (black dots in Fig. 11b), IBI-CCS_corr exhibits a freshening, whereas IBI-CCS raw does not show a particular change. This result confirms those of Sect. 3.1.3: bias corrections could lead to a change in the TS diagram shape. Although the bias corrections of temperature and salinity are stationary, the projected changes in the TS diagrams of the regional simulations with and without corrections are not the same and thus depend on the mean state. Bias corrections can therefore be important for projected water mass changes through the water column.

## 3.2.4 Projected changes of SL components

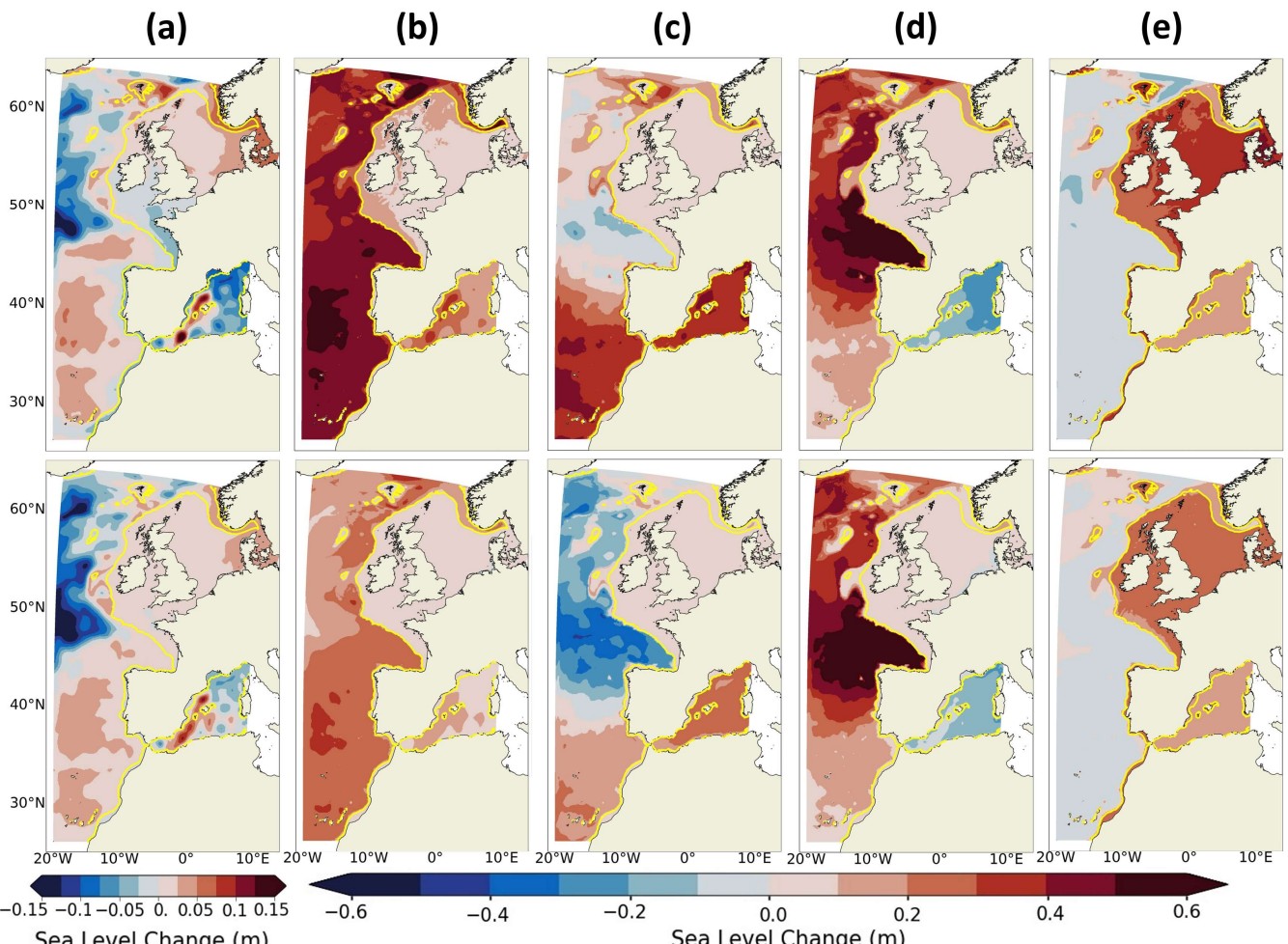

**Figure 12: Projected changes (2081-2100 vs 1986-2005) under the SSP5-8.5 (upper row) and SSP1-2.6 (lower row) scenarios in the IBI-CCS_corr simulation for the (a) DSL, (b) steric, (c) thermosteric (d) halosteric and (e) manometric SL components. The steric, thermosteric and halosteric SL components have been computed between 0-2000m depth. Note that the DSL mean over the IBI zone is 0. Note that the projected changes of the inverse barometer effect are not incorporated in the projected changes of manometric sea level (e). Moreover, as the changes are computed 0-2000m, the steric + manometric sea level changes are not equal to the DSL changes here. The shelf break (defined by the 200 m isobath is indicated in yellow).**

Figure 12 shows the projected changes of SL components described in Sect. 2.3 for SSP5-8.5 and SSP1-2.6 scenarios for the IBI-CCS_corr regional simulation. For all the SL components, spatial patterns of projected changes are quite similar under the two scenarios. Indeed, for the DSL changes, both scenarios exhibit an increase of the DSL in the North Sea and a decrease in the northwestern part of the domain (Fig. 12a). The main difference between the two scenarios is the projected slight decrease in the DSL in the Bay of Biscay under SSP5-8.5, which is not projected under the SSP1-2.6 scenario. Figure 12b shows a large steric SLR in the deep ocean compared to the shelf, as expected for this depth-integrated variable. This result is consistent with Fox-Kemper et al., 2021. The corresponding steric SL gradients are compensated by shelf mass loading (Richter et al., 2013). Indeed, Figure 12e shows a slight decrease of manometric SL in the deep ocean and a substantial manometric SLR over the shelf. In general, steric and manometric SL changes are of smaller amplitude under SSP1-2.6 than under SSP5-8.5. Thermosteric SL is projected to increase south of around 40°N and in the Mediterranean Sea, but to decrease in the deep ocean north of 40°N, especially under SSP1-2.6 (Fig. 12c). This pattern is inherited from the GCM forcing at the boundaries. A smaller projected decrease in thermosteric SL in the northwestern part of the domain has also been observed in Hermans et al., 2020b with the MPI-ESM-LR GCM. The projected warming in the Mediterranean Sea is consistent with Adloff et al., 2015 and Soto-Navarro et al., 2020. Both scenarios exhibit an increase of halosteric SL in the Atlantic Ocean, contrary to Hermans et al., 2020b, and a decrease in the Mediterranean Sea (Fig. 12d). For the Mediterranean Sea, it seems there is no clear feature

in projected changes in salinity in the different simulations from Soto-Navarro et al., 2020. Moreover, the global halosteric SL projections of Fox-Kemper et al., 2021 show a low model agreement over the IBI region based on 17 CMIP6 GCMs.

### 3.2.5 Impact of the resolution for the regional projections of SL

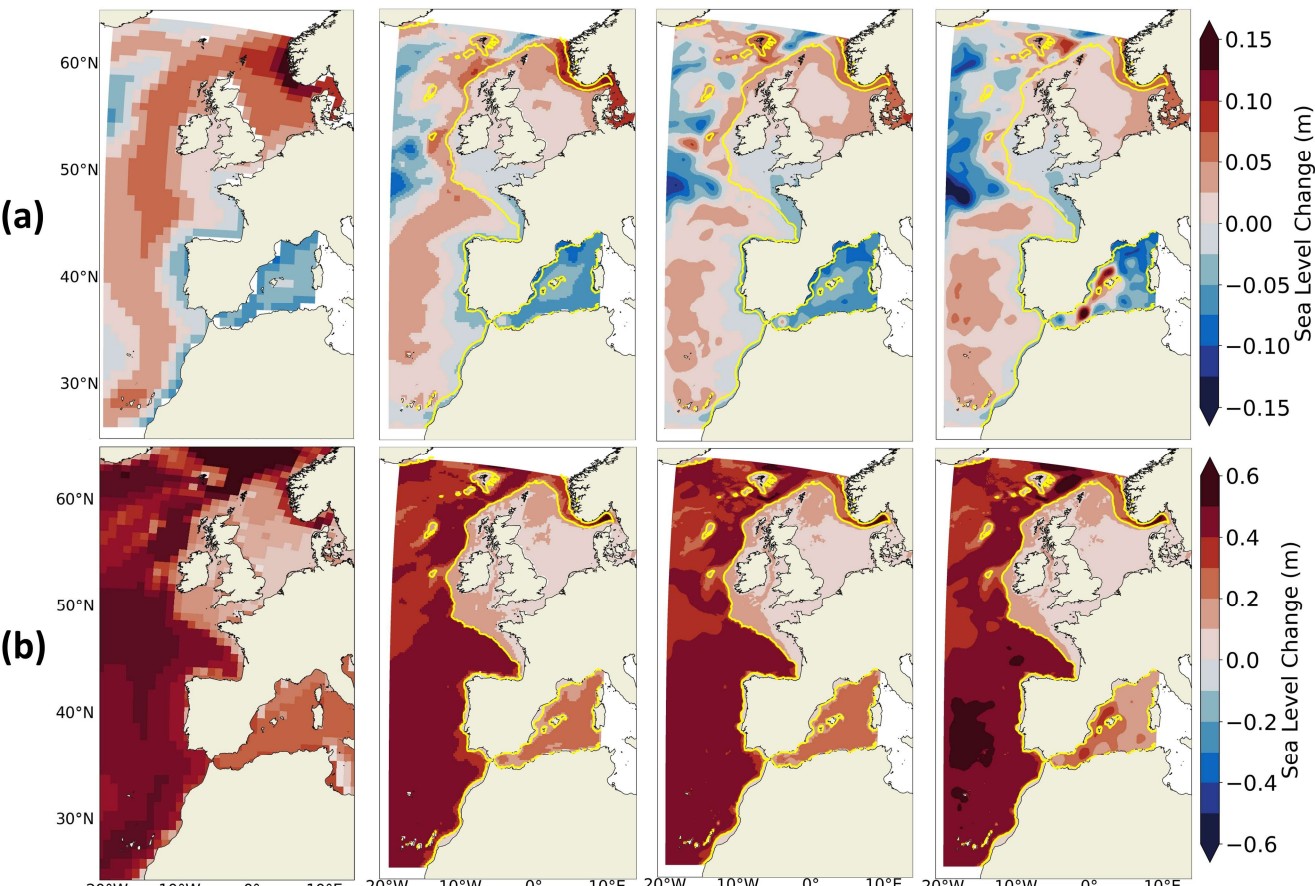

**Figure 13: Projected changes (2081-2100 vs 1986-2005) in (a) DSL and (b) steric SL over 0-2000 m depth under the SSP5-8.5 scenario in CNRM-CM6-1 (first column), CNRM-CM6-1-HR (second column), IBI-CCS_raw (third column) and IBI-CCS_corr (last column) simulations. Note that the DSL mean over the IBI zone is 0 in the RCMs and thus, to compare the DSL between the GCMs and RCMs, the mean DSL over the IBI domain is removed from the GCMs. The shelf break (defined by the 200 m isobath is indicated in yellow).**

Figure 13 compares the projected changes in DSL and steric SL in both GCMs and RCMs under the SSP5-8.5 scenario to assess the impact of the resolution on SL projected changes. The spatial patterns of steric SL projected changes are very similar for all the simulations (Fig. 13b) and agree with global projections from Fox-Kemper et al., 2021 and regional projections from Hermans et al., 2020b. For DSL changes (Fig. 13a), the different simulations are in good agreement with a projected increase of the DSL in the North and Baltic Seas, especially off the coasts of Scandinavia. Close to the western boundary, a decrease in DSL is projected in all simulations north of around 50° N (Fig. 13a). In the Mediterranean Sea, each simulation shows different projected DSL and steric SL changes (Fig. 13a,b). Globally, the spatial pattern of projected changes of the steric SL, and more importantly of DSL, in CNRM-CM6-1-HR and IBI-CCS have significantly more spatial information at the coast compared to the lower resolution CNRM-CM6-1. In addition, the added value of the high resolution GCM and regional simulations compared to the GCM CNRM-CM6-1 appears on the steric SL changes, where strong gradients of bathymetry are found.

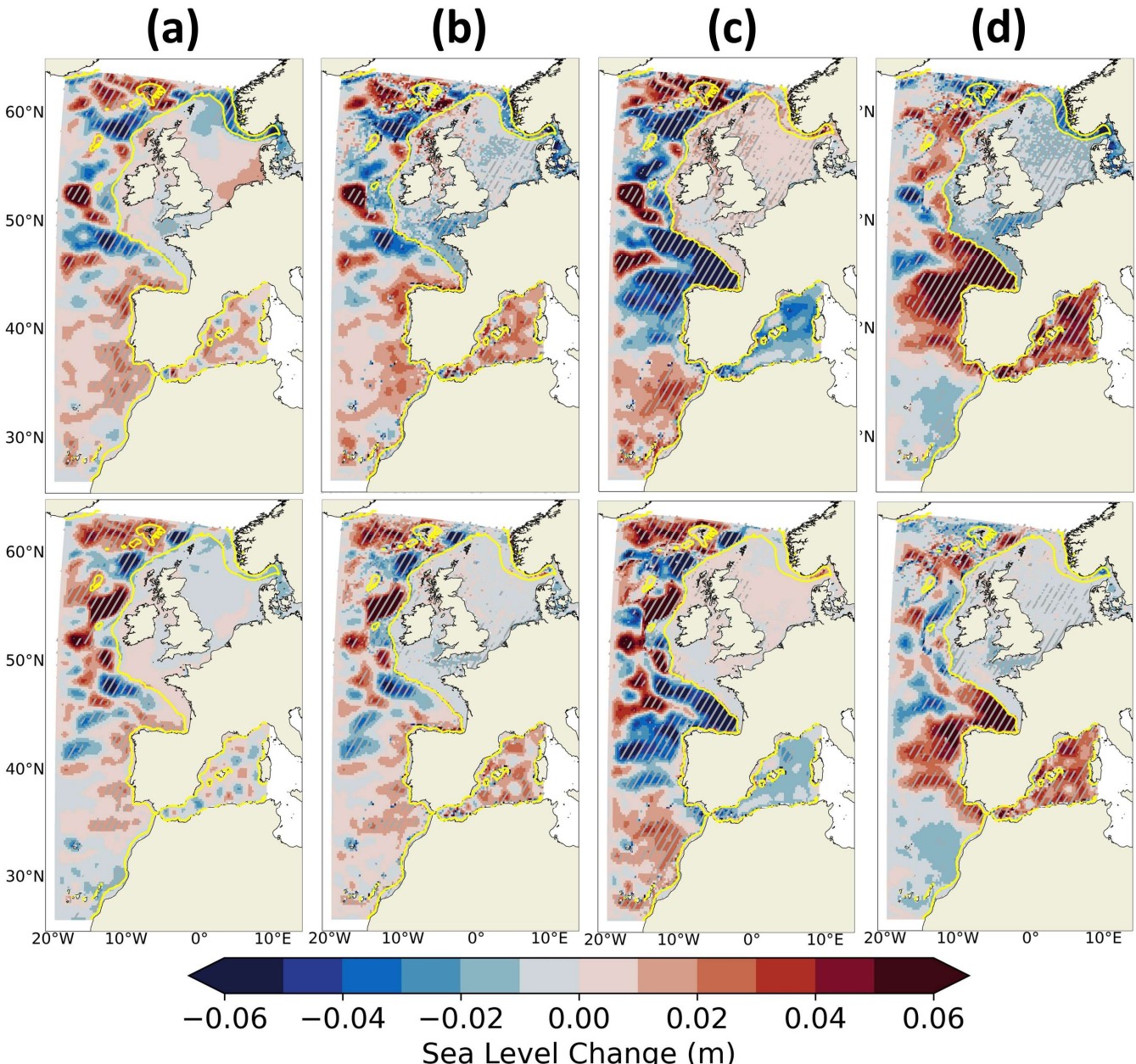

**(a)**        **(b)**        **(c)**        **(d)**

Sea Level Change (m)

**Figure 14: Differences of projected changes (2081-2100 vs 1986-2005) under the SSP5-8.5 (upper row) and SSP1-2.6 (lower row) scenarios between CNRM-CM6-1-HR and IBI-CCS_raw for the (a) DSL without IB effect, (b) steric, (c) thermosteric and (d) halosteric SL components. The steric, thermosteric and halosteric SL components have been computed between 0-2000 m depth. The differences have been computed on the GCM grid. A student test has been performed with a confidence interval of 95% and the significant differences between the two simulations have been marked with white dots. The shelf break (defined by the 200 m isobath is indicated in yellow).**

To isolate the impact of dynamical downscaling on the projected changes, Figure 14 shows the differences in SL drivers' projected changes between the regionally downscaled IBI-CCS_raw and the global CNRM-CM6-1-HR. The comparisons allow us to assess the impact of the increased model resolution, the different parametrizations, and the added processes for the projections of the different SL components. The consistency of the changes between the 2 scenarios suggests a robust climate change signal rather than a signal dominated by internal climate variability (Fig. 14). Differences in projected DSL changes due to a higher resolution are generally more important for the SSP5-8.5 than for the SSP1-2.6 scenario.

In coastal zones, the largest differences in projected DSL are found along the Norwegian coasts, with a 5cm smaller projected change in the IBI-CCS_raw simulation compared to the GCM under SSP5-8.5 (Fig. 14a). This difference is related to the strong decrease in the surface circulation of the Norwegian Coastal Current (Fig. 1) in the GCM but not in the RCM (Fig. 10).

Substantial differences in projected DSL changes are also found around Iberia (Fig. 14a) and are mostly related to differences in halosteric SL projected changes, which are partly compensated by differences in thermosteric projected changes (Fig. 14c,d). Otherwise, differences in projected changes in DSL between the GCM and IBI-CCS_raw are rather small in coastal areas, which is due to the relatively high resolution of the GCM and in particular of its bathymetry and land mask. This is consistent with the findings of Hermans et al., 2020b, where a larger impact of increased resolution through dynamic downscaling was found as they used coarser GCMs. They highlighted the importance of a realistic bathymetry and land mask for SL projections. Moreover, the impact of the higher resolution is rather small due to the peculiarities of the region, as the IBI zone includes many continental shelves. In shallow regions such as continental shelves, the Rossby radius is smaller than in the surrounding deep ocean, which requires an even higher resolution to resolve mesoscale processes. Over the northwestern European continental shelf, a resolution of at least 1/50 ° is required for ocean models to be eddy resolving, while models at 1/12 ° are eddy resolving in the deep part of the Atlantic domain in IBI (Hallberg, 2013). The RCM is therefore eddy-resolving in the deep Atlantic part of the domain, while the GCM is only eddy-permitting. The small differences in coastal steric, thermosteric and halosteric SL projected changes are consistent with the DSL changes at the coast (Fig. 14b,c,d). In the deep ocean and particularly in the northwestern part of the domain, where surface circulation changes are most important, differences between the GCM and RCM largely exceed the differences on the shelf for both scenarios and for all the SL components but their spatial patterns are rather noisy (Fig. 14).

### 3.2.6 Impact of bias corrections on regional projections of SL

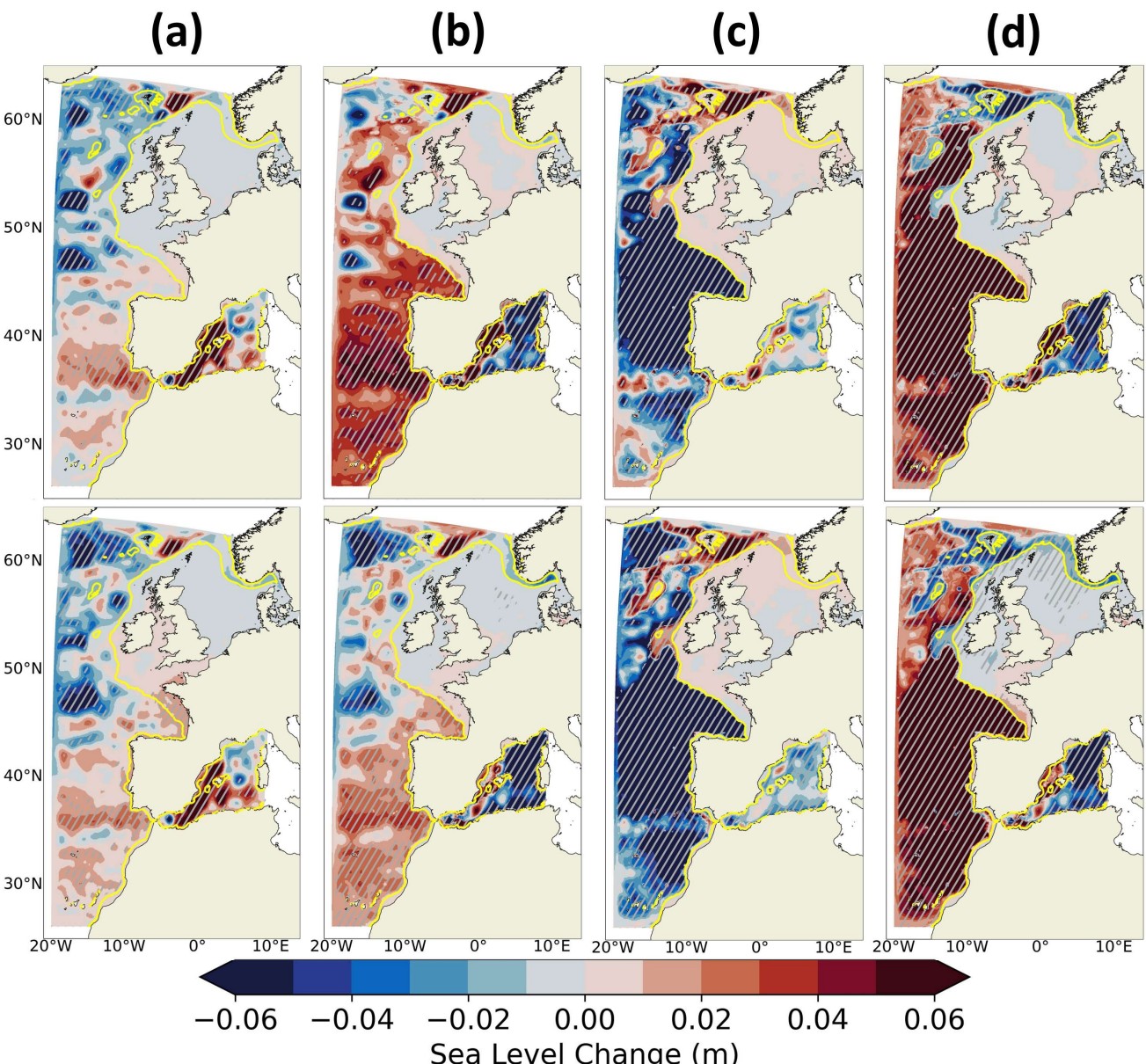

**Figure 15: Differences of projected changes (2081-2100 vs 1986-2005) between IBI-CCS_corr and IBI-CCS_raw simulations for the DSL (first column), the steric (second column), thermosteric (third column) and halosteric (last column) SL components under the SSP5-8.5 (upper row) and SSP1-2.6 (lower row) scenarios. The steric, thermosteric and halosteric SL components have been computed between 0-2000 m depth. A student test has been performed with a confidence interval of 95% and the significant differences between the two simulations have been marked with white dots. The shelf break (defined by the 200 m isobath is indicated in yellow).**

The impact of bias corrections on SL projections are now investigated by comparing projected SL changes between IBI-CCS_corr and IBI-CCS_raw (Table 1). The main spatial patterns of the differences between the projected changes of IBI-CCS_raw and IBI-CCS_corr are very similar for the two scenarios (Fig. 15), except in the Mediterranean Sea for the halosteric SL changes (Fig. 15d). Elsewhere, the impact of the corrections does not seem to be affected by the scenario. In general, large differences in projected SL changes between the two simulations are found in the deep ocean. However, the impact of bias corrections in coastal areas is small for both scenarios and for all the SL components.

This section focuses on the DSL changes (Fig. 15a). Differences in DSL changes in the deep ocean appear to be independent of the climate change scenario (Fig. 15a). In the northwestern part of the IBI domain, where the surface circulation changes are most important, the projected DSL changes are up to 10 cm smaller in IBI-CCS_corr (Fig. 15a). In the Mediterranean Sea,

where the bias corrections are substantial, differences in DSL changes are up to 15 cm larger in IBI-CCS_corr compared to IBI-CCS_raw in the Alboran Sea, which is associated with a larger increase in the net transport through Gibraltar Strait (not shown here). Also, the Alboran gyre is projected to decrease in IBI-CCS_corr under both climate change scenarios, whereas it is projected to strengthen in IBI-CCS_raw (Fig. 15a and Fig. 13a). The impact of bias corrections in coastal areas is rather small for the SSP5-8.5 scenario (except in the Mediterranean Sea) and larger for the SSP1-2.6 scenario (Fig. 15a). For instance,

the projected DSL changes in the SSP1-2.6 scenario are up to 2 cm larger in IBI-CCS_corr compared to IBI-CCS_raw in the Bay of Biscay and along the Iberian coasts, which is of similar amplitude to the projected DSL change in IBI-CCS_corr. Because of the stationarity of bias corrections, their impact on projected changes is larger when the climate change signal is smaller.

### 3.2.7 Projected changes of SL interannual variability

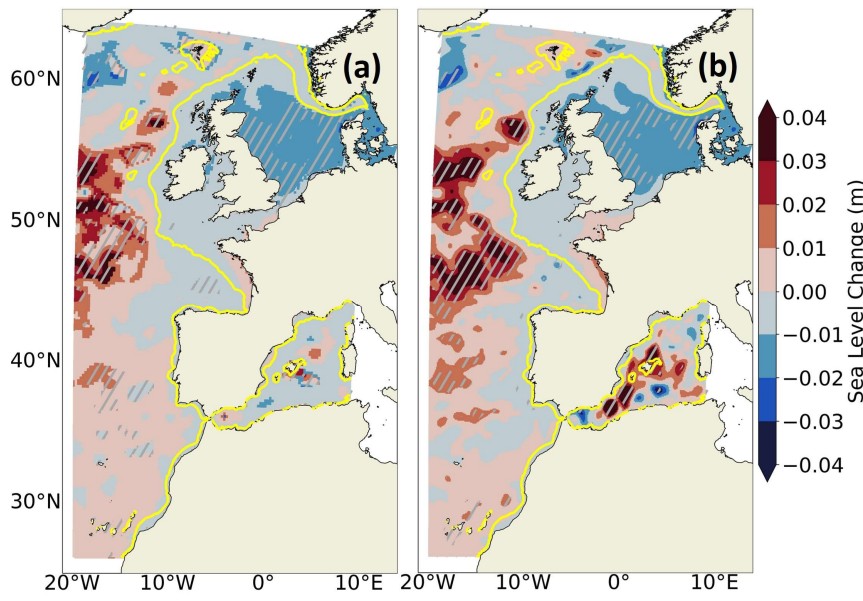


**Figure 16: Projected changes (2081-2100 vs 1986-2005) of SL interannual variability under the SSP5-8.5 scenario for the (a) GCM CNRM-CM6-1-HR and (b) IBI-CCS_corr. The interannual variability is computed as the standard deviation of the detrended annual mean SL. A Fisher test has been performed with a confidence interval of 90% and the significant changes between the two simulations have been marked with white dots. The shelf break (defined by the 200 m isobath is indicated in yellow).**

Figure 16 shows the projected changes in SL interannual variability under the SSP5-8.5 scenario for the GCM CNRM-CM6-1-HR and for the IBI-CCS_corr regional simulation. Thanks to the bias correction method used here, the internal variability of the GCM is conserved, which allows the projected changes of the variability in the regional simulations to be investigated. This would not have been possible with other correction methods, such as the delta method, i.e. mean state change projected anomalies added to historical forcings where the high frequency variability is, by design, unchanged between the global and

regional models.

Projected changes in interannual variability are consistent between the GCM and IBI-CCS_corr simulation (Fig. 16). Significant changes in the amplitude of interannual variability are observed where important changes in circulation are also projected (Sect. 3.2.2), e.g. the NAC and in the Mediterranean Sea. Projected changes are also significant in the North Sea

shelf in both GCM and IBI-CCS_corr. The decrease in the magnitude of the sea level interannual variability in the North Sea is associated with a reduction of the CNRM-CM6-1-HR wind variability over the same region (not shown), consistently with Hermans et al., 2020a, who have shown that the sea level interannual variability is mainly driven by the atmospheric forcing variability in the region. This change in wind forcing does not appear to be a robust climate change signal under the SSP5-8.5 scenario in the CMIP6 GCMs ensemble, thus the change in sea level variability in the North Sea is probably a specific feature

of CNRM-CM6-1-HR and IBI-CCS. Figure 16 displays changes in variability but it is not possible to state whether these changes are indeed reflecting changes in interannual variability or lower frequency signals such as multi-decadal variability.

### 3.2.8 Projected changes of extreme SLs

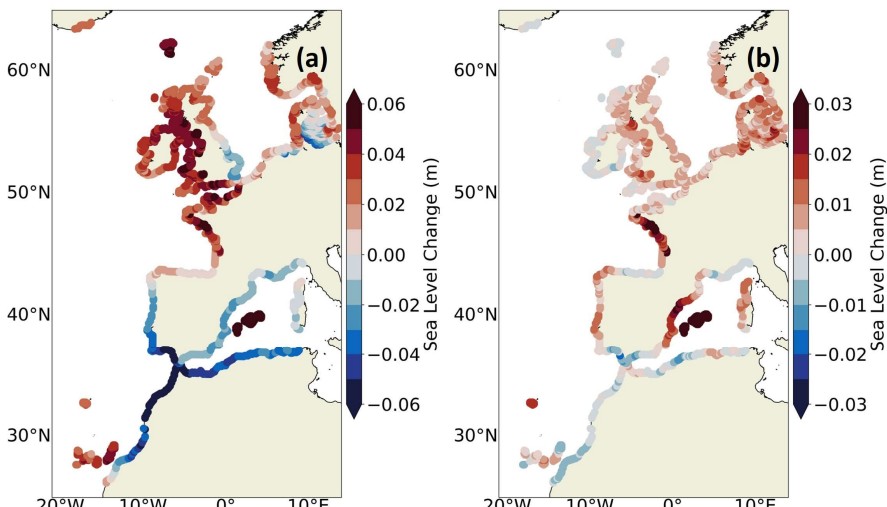

**Figure 17: (a) Projected changes of the non-tidal residual 99th percentile for the SSP5-8.5 scenario between 2081-2100 and 1986-**
**2005 period in IBI-CCS_corr. (b) Differences with projected changes in IBI-CCS_raw.**

For impact studies, it is necessary to consider projected SL extreme values. Figure 17 shows the projected changes in the 99th percentile of non-tidal residuals under the SSP5-8.5 scenario. Note that the mean has been subtracted on both the 1986-2005 and 2081-2100 time slices to remove the SLR effect on extreme SLs and to assess changes in the remaining component, which corresponds to the atmospheric surge. Changes in extreme SLs are therefore of a small amplitude of maximum 6 cm (Fig. 17a)
as changes in extreme SLs are mainly driven by SLR (Vousdoukas et al., 2018b; Muis et al., 2020). However, some spatial differences are found on the atmospheric surge with an increase in extreme SLs in the Armorican Shelf, English Channel, Celtic Sea, Irish Sea (Fig. 1). The impact of the bias correction on this high-frequency diagnostic is assessed in Fig. 17b. Projected changes of 99th percentile of non-tidal residuals are relatively weakly affected by the bias correction, except on the Armorican Shelf (Fig. 1) and in the Mediterranean Sea, where the differences with IBI-CCS_raw are half the climate change
signal.

### 3.2.9 Projected changes in the M2 tidal amplitude

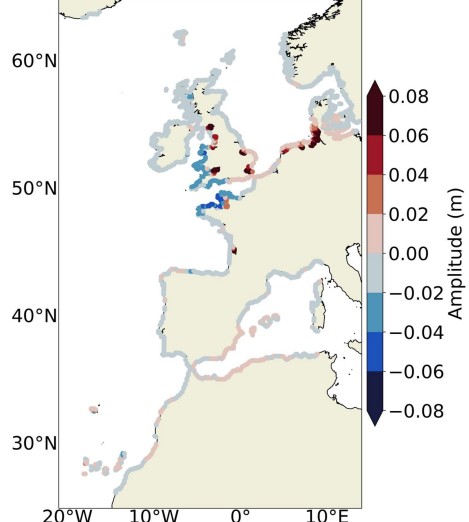

**Figure 18: Projected changes of the M2 tidal amplitude for the SSP5-8.5 scenario between 2081-2100 and 1986-2005 period in IBI-CCS_corr.**

The main added value of the RCM in comparison to the GCM is the inclusion of processes driving SL changes at the coast, such as tides (Sect. 2.1.2). Here, the projected changes of the major tidal constituent (M2) amplitude are assessed under the SSP5-8.5 scenario (Fig. 18). These changes in amplitude are important, as tides are major drivers of extreme SLs. Projected tidal changes over the 21$^{st}$ century are mostly due to SLR (Fox-Kemper et al., 2021). However, projected changes are not expected to be large in IBI-CCS_corr, as the tidal potential imposed at the boundaries of the domain does not change in

response to climate change and as coastlines are fixed in the model (no wetting and drying, erosion etc). Nevertheless, tides can be impacted by the SLR on the shelf, far enough from the boundaries. Here, with a SLR of about +80 cm at the end of the century (Sect. 3.2.1), the M2 tidal amplitude seems to be impacted mostly in the southern part of the North Sea on the large continental shelf. In this region, the M2 tidal amplitude is projected to increase by 10%, which is consistent with Idier et al., 2017. Projected changes in M2 phase were also assessed under the SSP5-8.5 scenario and show no difference compared to the

historical simulation and no displacement of the amphidromic points (not shown here).

## 4 Discussion and Conclusions

Previous dynamical downscaling studies have provided regional projections of SL based on low-resolution CMIP5 GCMs (e.g. Hermans et al., 2020b; Liu et al., 2016; Zhang et al., 2017; Gomis et al., 2016, Jin et al., 2021). The objective of this study was to present a regional ocean model, called IBI-CCS, to be used for analyzing sea level in the northeastern Atlantic region bordering western Europe, based on a 1/4° resolution CMIP6 GCM. To limit the GCM bias propagation into the regional

projections, seasonal mean bias corrections were applied to the GCM outputs (2D and 3D variables) prior to their use in the RCM. In this study, this configuration is presented along with its evaluation and the added value of the regional model vs the global model. Several sensitivity experiments are analyzed to disentangle the respective effects of dynamical downscaling and of applying bias corrections both on the current climate and on the projected change.


Comparisons between the GCM and the regional simulation without bias correction were performed to assess the impact on the simulations of the dynamical downscaling of a ¼° resolution GCM. More specifically, we investigated the influence on the simulations of: (1) the higher resolution and (2) the inclusion of processes driving SL at the coast in the RCM (tides and atmospheric surface pressure forcing). These comparisons show that the dynamical downscaling method conserves the GCM

spatial patterns, along with its interannual variability and trends. Over the historical period, the 1/12 ° resolution of the regional model IBI-CCS allows for better regional circulation and SL with more spatial information. Although the impact of the increased resolution on the deep ocean projections is significant, this is not the case for the coastal projections, where the impact of the increased resolution is limited. This is due to the relatively high resolution of the GCM associated with a quite realistic bathymetry and land mask. Additionally, the IBI zone contains a high concentration of continental shelves, where the

Rossby radius is small, which requires a resolution of at least 1/50 ° to be eddy resolving. This limits the gains expected over continental shelves in the IBI region from the RCM resolution of 1/12 °. We expect that the impact of dynamical downscaling would be much larger if the GCM had a more typical CMIP ocean model resolution of 1 °. Hermans et al., 2020b previously proposed this argument based on the downscaling of two GCMs with different ocean grid resolution in the region. The choice of the eddy-permitting high-resolution GCM allows a realistic regional circulation and MSSH to be obtained, which would

probably not have been the case with a 1 ° GCM as shown in the comparisons between CNRM-CM6-1-HR and CNRM-CM6-1. However, to answer this question definitively, it would be necessary to downscale CNRM-CM6-1 as well, which is beyond the scope of this study. Nevertheless, thanks to the physical processes included in the RCM and not taken into account in the GCM, such as tides and atmospheric pressure forcing, high frequency SLs variations are represented in more detail in the IBI-CCS simulations. The validation shown here provides some confidence in terms of the realism of the representation of these

processes and paves the way for a future analysis more focused on extreme SLs projected changes.

The effect of the bias correction has been clearly established: the large-scale performances of IBI-CCS are better than those of the GCM in terms of SL components, regional circulation and representation of water masses. For instance, the characteristics of the water masses were corrected, thereby controlling the influence of the initially highly biased Mediterranean Sea on the Atlantic Ocean. For the projected changes, the bias corrections have a significant impact on the deep ocean but less at the coast in general, except in the Mediterranean Sea, where the biases were substantial. Additionally, due to the stationarity of the bias corrections, their impact on the projected changes is larger for the SSP1-2.6 scenario, where the climate change signal is weaker and of comparable magnitude as the bias corrections than for the SSP5-8.5 scenario, where the climate signal dominates. This method appears to be applicable to other, even strongly biased, models. Moreover, although the corrections were stationary, the projections of the water mass properties in the simulations with and without bias correction were different for a given depth (different TS diagram shape). We applied a seasonal bias correction method that has been widely used in other studies for 2D variables (Adloff et al., 2015, 2018). The purpose of this paper was not to develop a new correction method, but rather to use a state-of-the-art method to develop our model configuration and simulations. However, this bias correction method assumes that biases are stationary, while several papers have shown the non-stationarity of biases (Maraun, 2012; Nahar et al., 2017; Hui et al., 2020). Another caveat is that 3D variables are independently corrected. However, the large amplitude of biases found in the GCM justifies the use of stationary seasonal mean bias corrections to address them. The bias correction method was chosen over the delta method (in which the mean climate change signal is added to the present-day time series) because it preserves the variability of the GCM. The internal variability of the regional model is thus driven by the GCM for the historical period and projections, allowing projected changes of the variability in the regional simulations to be investigated. More sophisticated bias correction methods are currently being developed (such as emergent constraints or multi-variable corrections). However, these methods are not yet mature enough and are potentially difficult to apply in the case of a 3D model.

The use of a single forcing GCM and a single member does not allow quantification of the uncertainties of the projected results. Here, the aim of the study was not to characterize the uncertainties nor provide a likely range of projected changes over the IBI region. Rather, the regional configuration was developed to investigate questions related to SL changes in the IBI region in terms of processes, not uncertainties. To gain insight into the representativeness of the GCM forcing model chosen here, we verified that the GCM was not an outlier of the CMIP6 models for a set of metrics relevant for SL changes on the IBI zone. In a way, the differences in the regional climate change projections with and without applying bias corrections are another indication of the uncertainties of the modelling chain when analyzing regional climate simulations (Hernández-Díaz et al., 2019).

The methodology used here to produce climate projections dealing with GCM biases is intended to be applied to a larger number of models in forthcoming studies. However, when considering a large number of models, the bias correction method may not be systematically applicable. Moreover, the model set-up requires more computational effort than time-slice methods such as that used by Jin et al. (2021), which makes it difficult to obtain the large ensembles that are eventually required for comprehensive projections. In the case of long simulations, it may be preferable to select the forcing models based on one or several criteria before using them for projections. The best would be to eliminate the models that have strong difficulties in the area considered and for the key variables of the intended study. Emergent constraint methods were also developed to overcome model biases and better characterize the uncertainties of the projections (Chen et al., 2020; Grinsted and Christensen, 2021; Forster et al., 2021).

In conclusion, in this paper the IBI-CCS regional model appears to be a suitable tool for investigating questions related to climate change over the ocean in the IBI region, especially regarding SL. The aim of a follow-up study will be to analyze the

projected changes in extreme SLs. For this purpose, the model will be further validated on extreme SLs; for example, in terms of return periods and return levels.

**Code availability**

The IBI-CCS model is based on the NEMO 3.6 version developed by the NEMO consortium. All specificities included in the

NEMO code version 3.6 are freely available (https://www.nemo-ocean.eu/).

**Data availability**

Information on CNRM-CM6-1-HR and CNRM-CM6-1 simulations can be found at https://doi.org/10.22033/ESGF/CMIP6.4067 (CNRM-CM6-1-HR, historical), https://doi.org/10.22033/ESGF/CMIP6.4164 (CNRM-CM6-1-HR, piControl), https://doi.org/10.22033/ESGF/CMIP6.4185 (CNRM-CM6-1-HR, ssp126),

https://doi.org/10.22033/ESGF/CMIP6.4225 (CNRM-CM6-1-HR, ssp585) and https://doi.org/10.22033/ESGF/CMIP6.4066 (CNRM-CM6-1, historical). The CNRM-CM6-1-HR forcing fields are available on the ESGF website (https://esgf-node.ipsl.upmc.fr/projects/esgf-ipsl/). The reanalyses data and altimetric observation product were obtained from the Copernicus Marine Services (https://marine.copernicus.eu/). MDT CNES CLS18 was produced by CLS and distributed by Aviso +, with support from Cnes (https://www.aviso.altimetry.fr/). TG data records are obtained from the GESLA dataset

(https://www.gesla.org/)

**Supplement link**

**Author contribution:**

AM designed the study. AV performed the global simulations. GR prepared the regional model configuration. AC prepared the forcing files, performed the regional simulations and did the analyses. AM, AV and GR supervised the project. AC wrote

the first draft of the manuscript. All authors contributed to manuscript revisions and read and approved the submitted version.

**Competing interests**:
All authors declare that they have no conflicts of interest.

**Disclaimer**

**Acknowledgements**

Analyses were carried out with Python. The authors thank Romain Bourdallé-Badie for his useful technical advice on regional modeling.

**Financial support**

The PhD thesis of AC is supported by Mercator Ocean and Météo-France.

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
