# Peer review of "IBI-CCS: a regional high-resolution model to simulate sea level in Western Europe"

_Geoscientific Model Development, 2021_

## Author Comment (AC1)

08.02.2022

Answer RC1:

The authors wish to thank the anonymous reviewer for his/her detailed and constructive review. The comments, questions and remarks greatly helped us improving the quality of the paper. We are pleased to address you point-by-point answers to your review in blue in the response below.

Best regards,

The authors.

**Main comments:**

L86 states that the aim of the study is to provide projections of sea-level changes, focusing on methodological aspects. If this is the aim, using only a single (downscaled) GCM is probably insufficient. The specification in L86-L90 suggests, however, that the aim is to evaluate the impact of dynamical downscaling and bias corrections on simulations of sea-level change, while the title suggests that the aim is to 'evaluate sea-level change'. Other statements in the manuscript imply that the study is meant for presenting and evaluating a regional ocean model that will apparently be used for analyzing extreme sea level in a follow-up study. Altogether, I think that the purpose of the manuscript needs to be more clearly described in the introduction, in the conclusions and possibly in the title as well.

We thank the reviewer for this comment. The study is meant to present a regional ocean model that will be used for analyzing the sensitivity of sea level changes, particularly extreme sea level changes, to methodological choices and representation of processes. In the current study, the configuration is presented along with its evaluation and the added value of the regional vs global model. It is from this perspective that we assess the impact of the dynamical downscaling and bias corrections on simulations of sea level changes. As the aim of the manuscript is to present the ocean regional model even more than to evaluate it, the title has been changed to "IBI-CCS: a regional high-resolution model to simulate sea level in Western Europe". The methodology presented in this paper could subsequently be applied to produce an ensemble of simulations using different CMIP6 global models as parent models to provide projections of sea level changes and related uncertainties. The introduction and conclusion have been revised to clarify the aim of the manuscript. In L86, the term "focusing on methodological aspects" was referring to the dynamical downscaling methodology and bias corrections applied and not to the uncertainties associated with the simulations. The term has been removed to avoid confusion.

L221-L239: the authors apply a bias correction to the boundary conditions by subtracting the historical mean seasonal cycle of biases, assuming that biases are stationary. However, the seasonal cycle and therefore the associated model biases are likely to change in the future. Could the authors comment on the validity/caveats of their methodology in this light? Showing the size of the bias (corrections) may be insightful as well.

We applied a seasonal bias correction method that has been widely used in other studies for 2D variables (Adloff et al., 2015, 2018). The purpose of this paper was not to develop a new correction method, but rather to a use state-of-the-art method to develop our model configuration and simulations. However, this bias correction method assumes that biases are stationary, while several papers have shown the non-stationarity of biases (Maraun, 2012; Nahar et al., 2017; Hui et al., 2020). Another caveat is that 3D variables are independently corrected. However, the large amplitude of biases found in the GCM justifies the use of stationary seasonal mean bias corrections to address them. The bias correction method was chosen over the delta method (in which the mean climate change signal is added to the present-day time series) because it preserves the variability of the GCM. More sophisticated bias correction methods are currently being developed (such as emergent constraints or multi-variable corrections). However, these methods are not yet mature enough and are potentially difficult to apply in the case of a 3D model. This discussion on bias correction methods is now reflected in the Discussion and Conclusions section of the manuscript.

Figure RC1 shows the amplitude of the temperature and salinity bias corrections for the western boundary of the IBI domain. The amplitude of the biases is very large and can reach up to 4.5 ° and 1.5 psu at 1500m depth. At 1500m depth, the biases of the western boundary are directly related to the Mediterranean Sea biases, mostly due to the Rhone river runoff error in the GCM. Their large amplitude justified for us the corrections applied to the IBI-CCS_corr simulation for the runoffs and thus also for the boundaries for consistency reasons. The temperature bias has a strong seasonality until 200m depth (Fig. RC1a and c) which is why we decided to apply a seasonal mean bias correction. Figure RC1 has been added to the Supplementary Materials.

[Figure]

**Figure RC1: Temperature bias over the 1993-2014 period between CNRM-CM6-1-HR and GLORYS for the winter (a) and for the summer (c). Salinity bias over the 1993-2014 period between CNRM-CM6-1-HR and GLORYS for the winter (b) and for the summer (d). The biases are computed over the western boundary of the domain at 20°W.**

In addition, Figure RC2 shows the seasonal sea surface temperature (SST) bias of CNRM-CM6-1-HR over the IBI domain using different reference datasets and over different periods: over 1993-2014 using the IBIRYS reanalysis (Levier et al., 2020), the GREP ensemble of reanalyses (Desportes et al., 2017), the observations CORA5.1 (Szekely and Roltan 2020) and ARMORD3D (Guinehut et al., 2012) as reference datasets; over 1935-1978 and 1900-1992 (the two available periods) using LEVITUS (Levitus 1982) as a reference dataset. The seasonal bias appeared to be quite stationary for the different past periods. For instance, the amplitude of the seasonal bias - 2°C - is 4 times larger than the amplitude of the difference of monthly biases for different periods / reference datasets - 0,5°C -.

[Figure]

**Figure RC2: Seasonal sea surface temperature (SST) bias of CNRM-CM6-1-HR over the IBI domain using different reference datasets and over different periods.**

Moreover, Figure RC3 shows the seasonal SST projections of CNRM-CM6-1-HR over the IBI domain. In projections, the GCM does not exhibit a shift in the seasonal cycle but only an offset in SST amplitudes, so the seasonal mean bias corrections did not seem detrimental to us.

[Figure]

**Figure RC3: The seasonal SST projections (2081-2100) of CNRM-CM6-1-HR over the IBI domain for different climate change scenarios in comparison the 1986-2005 period.**

Section 3.1: In multiple comparisons in this section, the IBIRYS and IBI-ERAi products are taken as the ground truth for evaluating the added value of downscaling, while those products rely on models and have their own biases as well. A quantitative comparison against observations to support the case that the changes due to downscaling and bias corrections are actually improvements, seems to be missing.

It is true that the models have their own biases but there is also a lack of 4D observations which limits the validation of integrated fields such as thermosteric, halosteric and steric sea level. The paragraph of Sect. 2.1.2 has been revised: "The IBI zone is covered in the framework of the Copernicus Marine Service (CMEMS) with a 1/12 ° reanalysis IBIRYS (https://marine.copernicus.eu/about/producers/ibi-mfc) including assimilation of observations and bias corrections. The configuration of IBIRYS (described in Baladrón et al., 2020 and validated in Levier et al., 2020) was used to perform the IBI-CCS simulations. The reanalysis has shown very good performances against observations in Levier et al. 2020 for several variables used to validate the IBI-CCS simulations in Sect. 3.1: temperature, salinity, currents, sea level. As IBIRYS and IBI-CSS share the same modeling framework, better results are not expected with IBI-CCS_corr than with IBIRYS. Therefore, IBIRYS is considered as the reference for the IBI zone in the study."

In addition, some comparisons against observations have been performed in Sect. 3.1 for the sea level metrics: the mean sea surface is compared to the mean dynamic topography based on observations (altimetry, gravimetry and in situ data), the modeled sea level interannual variability is compared to a global gridded reprocessed altimetric observation product distributed by CMEMS, the non-tidal residuals are compared to tide gauge data.

Section 3.1.6: the authors compare the 99th percentile SSH between IBI-CCS_corr and GESLA2 observations and conclude that their model 'properly' reproduces the observed extremes, with an 'error rarely exceeding 20%'. What is missing, however, is their motivation and contextualization for why errors of 10-20% are acceptable. Additionally, the authors should motivate why assessing only this single aspect of the simulation of ESLs is sufficient evidence that IBI-CCS is 'a suitable tool (L790)' for analyzing extreme sea levels and projecting their changes, as the authors seem to plan doing in a follow-up study.

Thank you for your remark, the sentence has been revised in Sect. 3.1.6: "Errors found between IBI-CCS simulations and GESLA dataset are comparable to those of recent papers (Muis et al., 2020; Kirezci et al., 2020)."

Then, only this single aspect of the simulation of ESLs is not sufficient and a more complete validation of the ESLs will be done in the follow-up study. L529, a sentence has been added: "The model will be further validated on extreme SLs; for example, in terms of return periods and return levels in a follow-up study." The conclusion of the last paragraph L790-792 has been moderated as follows: "In conclusion, in this paper the IBI-CCS regional model appears to be a suitable tool for investigating questions related to climate change over the ocean in the IBI region, especially regarding SL. The aim of a follow-up study will be to analyze the projected changes in extreme SLs. For this purpose, the model will be further validated on extreme SLs; for example, in terms of return periods and return levels."

**Other comments:**

L11: "some relevant processes' - here and elsewhere, the paper would benefit from the authors more precisely formulating what they mean

Here, the "relevant processes" are the processes driving coastal sea level changes which are not included in the GCMs: tides, atmospheric surface pressure forcing, waves.

L15: suggest changing 'developed for' to 'participating in'

Done.

L36: here and throughout, it would be good to avoid complicated compounds such as 'decision-making processes' (consider 'decision making'), 'SL focus' ('focus on sea level'), 'the model spatial resolution' ('the spatial resolution of the model'), SL changes spatial variations ('spatial variations of SLC'), etc.

Done.

L45: remove 'to' or replace by 'due to'

Done.

L51: could the authors add an average kilometric resolution? (only few CMIP6 models have a quarter degree resolution)

Done.

L56: The authors may consider writing out less commonly used abbreviations like DD, as well as the regional features in Figure 1, as these abbreviations may confuse readers unfamiliar with these terms

Abbreviations have been removed except for regional climate model (RCM), global climate model (GCM), sea level (SL), dynamic sea level (DSL), global mean sea level (GMSL) and global mean thermosteric sea level (GMTSL).

L60: 'thanks to DD' -> 'using DD'?

Done.

L75: is it dynamical downscaling that can overcome this problem or is it the bias corrections?

The sentence has been modified as follows: "To overcome this problem, bias corrections can be applied to the GCM outputs before using them as forcing when performing dynamical downscaling"

L81-82: unclear sentence structure

The sentence has been modified as follows: "Other methods exist, such as rescaling the data with a factor (Macias et al., 2018 on the winds) or individually adjusting different ranges of a distribution."

L85: up to this point the authors have made clear what has been done in the literature, but they have not pointed out what has not been done in the literature and therefore the motivation for and the novelty of their model/study is not yet clear.

A paragraph has been added L71: "Most of these studies have used low-resolution GCMs (around 1 ° spatial resolution), which explains the large differences in the projected sea level changes between GCMs and RCMs. However, few studies have employed higher resolution GCMs (e.g. ¼ ° over the ocean ≈ 28 km at the equator) in a dynamical downscaling framework. Hermans et al., 2020b have downscaled two GCMs with different ocean grid resolution. They concluded that the impact of dynamical downscaling is expected to be larger if the GCM has a lower resolution (typical CMIP5/6 model resolution of 1 °). The use of a GCM with a higher spatial resolution is nevertheless interesting for both the ocean (¼ °, eddy-permitting) and atmosphere. In particular, the atmospheric forcing applied to the regional ocean model is of higher resolution, which increases the realism of the forcing. For instance, the intensity of the atmospheric low-pressure systems and the spatial patterns generating extreme SL episodes should be better reproduced."

The sentence L77-79 has been modified: "This method is used with a seasonal bias correction on the sea surface temperature (SST) in Adloff et al., 2015 and seasonal bias correction on the sea surface height (SSH) in Adloff et al., 2018, but has never been applied to 3D variables in the case of a dynamic downscaling method."

L89: the adjective 'high-resolution' is somewhat misleading here: the resolution may be high compared to other CMIP6 GCMs, but is not that high compared to some regional ocean models. If it would be, there would be less reason to dynamically downscale.

The adjective "high-resolution" has been changed in "1/4° resolution".

L89: 'to these aims'

"To these aims" has been removed.

L99: I suggest to change 'climate change scenario' to 'greenhouse gas concentration scenarios' and add a reference. Additionally, the choice to use these two scenarios needs motivation.

The choice of the scenario has been more clearly highlighted in L99 : "Sect. 3.2 shows the RCM and GCM projections over the 21st century for two extreme greenhouse gas concentration scenarios of CNRM-CM6-1-HR : SSP5-8.5 and SSP1-2.6 (O'Neill et al., 2016)."

Further justification is provided in Sect. 2.1.1: "In this paper, due to the computational cost of the simulations, we focus on only two greenhouse gas concentration scenarios. To obtain the most contrasting results possible, the two extreme scenarios of CNRM-CM6-1-HR included in Tier 1 of ScenarioMIP were chosen: SSP5-8.5 and SSP1-2.6 with respectively a very high and low radiative forcing by the end of the century (O'Neill et al., 2016)"

L104: 'regional ocean climate model' change to 'regional ocean model'?

Done.

L103-L106: could the authors clarify why CNRM-CM6-1-HR specifically is chosen?

A paragraph has been added L105 to clarify the choice of the GCM: "CNRM-CM6-1-HR was chosen mostly for its high resolution over the ocean (0.25°, eddy-permitting) compared to the typical ocean grid resolution of CMIP6 models (1°). Other reasons for the choice of CNRM-CM6-1-HR are its high resolution over the atmosphere (0.5°) and its high frequency outputs. These two aspects are very important for the modeling of extreme SLs, which will be the purpose of a future study."

L111: does this mean that only the horizontal ocean resolution is increased in IBI-CCS compared to the CNRM-CM6-1-HR model?

Yes, exactly. The vertical ocean resolution is the same in IBI-CCS and CNRM-CM6-1-HR. A sentence has been added L111 to clarify it: "Only the horizontal ocean resolution is increased in IBI-CCS compared to CNRM-CM6-1-HR."

L119: 'to CMIP6 typical resolution' -> 'to the typical ocean grid resolution of CMIP6 models'

Done.

L130: 'required to force' -> 'for'

Done.

L135: 'approximatively' -> 'approximately'

Done.

L136-137: needs a reference

The reference Lee et al., 2021 has been added.

L149-155: I suggest to change the order: first introduce the grid (resolution) of the IBI-CCS model and then explain its grid is based on an existing reanalysis

Done.

L165: perhaps 'added value' here should be replaced by 'expected areas of improvement' or alike, since the added value is assessed later on in the manuscript

"The main added value" has been replaced by "The main improvement".

L188: the reason for the different set-up at the eastern boundary is missing

This is a historical setting that is certainly no longer relevant and does not deserve to appear in the manuscript. For the next simulations, this constraint will probably be removed. Therefore, the information has been removed from the manuscript.

L189: could the authors specify the type of boundary constraints used for the other variables as well?

The sentence has been modified L187: "Temperature, salinity and baroclinic velocities of the GCM are prescribed at the frontier point of the domain and a buffer zone of 10 grid points relaxes the internal regional solution to the prescribed boundary values."

L190-191: from where are the initial conditions and the runoff derived? From the GCM? Observations? elsewhere?

The initial conditions and runoff are derived from the GCM. It should be clearer thanks to the merging of Section 2.2.1 and the text above Table 1 as suggested in the following comment.

Section 2.2.1: could this usefully be merged with the text above table 1, leaving just a separate section describing the version with drift/bias corrections?

Done.

L211: could the authors specify how large? Does this have any effect on the SSP runs?

Figure RC4 shows the drift in 2100 for the temperature and salinity in the GCM (computed as the linear fit of the pre-industrial control run over the whole period 1850-2150) for the western boundary of the domain. In the red box (21°W, 57°N-60°N, 0-10m depth), the drift is of almost 1°C in 2100 (vs 1850) when the climate change signal is of +2.5°C between 1986-2005 and 2081-2100 for the SSP5-8.5 scenario. Therefore, the drift has an effect on the SSP runs especially for low greenhouse gas concentration scenarios (like the SSP1-2.6 scenario where the climate change signal is lower). That is why we had to correct it. A sentence has been added L212.

[Figure]

**Figure RC4: Drift in 2100 for the temperature (a) and salinity (b) for the western boundary of the IBI domain. Projected changes in temperature (c) and salinity (d) between 1986-2005 and 2081-2100 for the SSP5-8.5 scenario (not corrected from the drift) for the western boundary of the IBI domain.**

L215: please specify which variables, and could the authors explain how they know a linear fit is 'indeed appropriate'?

The variables concerned by drift are summarized in Table 1. For the ocean, the 3D variables concerned are the temperature, salinity and sea surface height and for the atmosphere, the 2D variables concerned are the 2m-air surface temperature and 2m-specific humidity. A linear fit is a widely used method to remove the drift (Gupta et al., 2013; Irving et al., 2021). For example, in Hermans et al., 2021, the drift for the variable "zostoga" appears nearly linear for most CMIP6 models. In the AR6 (Supp Mat Chap 9, section 9.SM.4.2), the variables "zos" and "zostoga" have also been corrected from the drift linearly. Also, it has been shown in other studies that removing a linear drift is a good compromise to avoid removing internal variability (Gupta et al., 2013).

A time series of the pre-industrial control run and its linear fit for temperature in the red box of the previous comment is provided in Figure RC5. In our case, the times series shows that a linear fit is appropriate to evaluate the drift.

[Figure]

**Figure RC5: Time series of the pre-industrial control run (piControl) and its linear fit for temperature at the western boundary of the domain (box: 21°W, 57°N-60°N, 0-10m depth).**

L216: subtracted 'from'

Done.

L269: The structure of the manuscript may benefit from a more descriptive title for this section

The title of Section 2.3.1 has been modified as follows: "Correction of the main SL components in the regional simulations".

L270-272: I think this line of reasoning could be clarified by writing that the frequency shift of ESLs depends on total sea-level rise rather than just the ocean dynamic component, hence additional SLC components need to be incorporated in the model.

Done.

L273: the contribution of changes in land-water storage to GMSLR seems to be neglected?

The contribution of changes in land-water storage to GMSLR is accounted for in CNRM-CM6-1-HR and IBI-CCS but not the associated GRD effects (Gregory et al., 2019). However, the contribution of changes is not evaluated nor corrected in the paper as they remain very small compared to the GMTSLR and to the contribution of changes in the cryosphere. The title of Section 2.3.1 has been modified as follows: "Transfer of water mass from the cryosphere to the ocean".

Section 2.3.3: it would help to clearly distinguish between sea level and sea-level change in the manuscript. For instance, is defined as total sea level (not sea-level change) but also as the sum of GMSL"R" and SSH. Sentence structure and grammar also need attention in this section.

The GMTSLR of the equation has been replaced by GMTSL. This section has been completely revised in order to distinguish the total sea level in both the global and regional models.

Figure 3: this figure shows many different things. To clarify things, the authors could consider splitting it up in multiple figures and discussing them one by one. I am also wondering, if the aim here is to evaluate depth-integrated salinity and heat, why these variables are then not evaluated directly, since their biases may be more intuitive to understand than their meter sea level equivalents. Also, in all maps in the manuscript, it would be helpful to indicate some of the relevant isobaths in the regions (e.g., the shelf break).

Indeed, Figure 3 is large and shows different things. However, the matrix presentation of Figure 3 had for us a real interest for the reading. The comparison of the rows gives information about the validation of the sea level components for a given simulation. The comparison of the columns provides information about the different biases between simulations for a given sea level component. In addition, the paper is already very long (35 pages) with many figures (18 figures) so we think that it is perhaps preferable to not split this figure.

The configuration has been set up to focus on the sea level and therefore we have chosen the sea level equivalent variables (steric, thermosteric and halosteric sea level) in this direction even if their biases may be less intuitive to understand than those for salt and temperature. The shelf break has been indicated in the maps.

Figure 4: apart from panel c, the arrows depicting the currents are too small to recognize

The figure has been modified with fewer arrows.

Figure 6: it would be helpful to mention that (presumably?) all panels here do not include the inverse barometer effect on mean sea level (same for Fig7?). Additionally, can the authors point to any reason for the differences in MDT between (a) and (b), especially in the Celtic Sea?

Thank you for pointing this out. A sentence is added in the captions of Fig. 6 and 7 of the manuscript: "Note that all panels here do not include the inverse barometer effect on sea level." The differences between (a) and (b) can be explained by the fact that IBI-ERAi does not include data assimilation. Therefore, the IBI-ERAi simulation has some biases compared to the observations in particular a negative bias in the Celtic Sea.

L462: the authors mention that all models reproduce the observed MDT (6a) 'well'. This assessment needs further justification given the differences between modelled and observed MDT in Figure 6.

The sentence L642 mentioning that all models reproduce the observed MDT "well" was meant for the Atlantic MSSH in the deep ocean and more specifically the northwest to southeast gradient associated to the NAC. To clarify, the sentence is modified as follows: "The simulations reproduced well the Atlantic main features of the observed MDT (Fig. 6a)."

L572-574: can the authors explain why it is expected that IBI-CCS_corr matches better with projections from CMIP(5?) models than IBI-CCS_raw? One may expect this to be the other way around, assuming that the CMIP models referred to also have deficiencies in resolving the Mediterranean Sea?

An explanation has been added L572-574: "In Soto-Navarro et al., 2020, the outlier CMIP5 simulations were excluded from the average T/S computation. The simulations accounted for are those with a good representation of the Mediterranean Sea. This is not the case of the GCM CNRM-CM6-1-HR used here to force IBI-CCS_raw which is probably an outlier simulation in this region. On the contrary, IBI-CCS_corr provides a much better representation of the Mediterranean Sea, which explains why the projected changes in IBI-CCS_corr are closer to the projections of Soto-Navarro et al., 2020 than those in IBI-CCS_raw simulation."

Figure 12: can the authors explain why they only computed steric changes 0-2000 m, and what the implications are for comparing results against studies that have integrated down to the full depth of the ocean? Does this mean that manometric + steric change in this case is not equal to total DSLC, since steric changes below 2000 m depth are missing? If so, that should be mentioned. Also, is the change of the inverse barometer effect incorporated in the manometric change here?

The steric changes have been computed 0-2000 m for the validation of the simulations. This choice has been made because the reanalyses include Argo observations which are provided up to 2000m. We chose to show the projected changes for the same 0-2000 m depth. Here, the manometric + steric change is thus not equal to the total DSLC for this reason but more importantly because we have set the spatial mean of DSLC to 0 over the IBI zone. This

has been done to focus on the spatial patterns of DSLC and to compare these patterns for the two GCMs (CNRM-CM6-1-HR and CNRM-CM6-1) and the RCM (IBI-CCS) in Figure 13 of the manuscript.

A sentence has been included in the caption: "Note that the DSL mean over the IBI zone is 0. Note that the projected changes of the inverse barometer effect are not incorporated in the projected changes of manometric sea level (e). Moreover, as the changes are computed 0-2000m, the steric + manometric sea level changes are not equal to the DSL changes here."

L593-611: I do not fully understand the sign change of thermosteric SLC in large parts of the Atlantic between SSP1-2.6 and SSP5-8.5. Additionally, the positive halosteric SLC in the Atlantic implies the whole 0-2000 m is getting fresher everywhere in the deep ocean. Can the authors point to any reasons for these effects? Have columns c and d of Figure 12 been interchanged accidentally, by any chance?

Column c and d have not been interchanged. The negative sign of thermosteric SLC is inherited from the GCM trough the boundaries. In this region, the GCM projects a cooling between 600-2000m depth and a freshening over the whole water column. In Fox-Kemper et al., 2021, Figure 9.12 shows in this zone a low CMIP6 model agreement (where <80% of the 17 GCMs agree on the sign of change) especially for the halosteric changes.

A possible explanation would be that this cooling is related to changes in the exchanges at Gibraltar Strait. Figure 11b in the paper shows that at the star location of Fig. 1 (and not in the Bay of Biscay, it has been corrected) the projected Mediterranean waters exit through the strait at a shallower depth than during the historical period. The "warm and salty" Mediterranean waters found in the Bay of Biscay in the past decades at ~800m depth (Figure RC6) are thus projected at a shallower depth too and therefore on thinner layers. For the same depth, between 600m depth to 2000m depth, the projected waters in the BoB are projected to be less warm and less salty. This explanation has not been included in the paper because it doesn't seem robust enough at this time.

[Figure]

**Figure RC6: TS diagram for the SSP5-8.5 scenario in the Bay of Biscay (box: 10°W-120W, 45°N-47°N) for the present (1986–2005) (light red) and future (2081–2100) (dark red) periods in IBI-CCS_corr.**

Figures 14 & 15: the white dots do not strongly contrast the shading of the figures, and in many cases cover most of the domain. I suggest choosing another color and only stippling those parts that have insignificant differences.

Figures 14,15 and 16 have been modified considering your remark.

L634: some caution is needed in attributing all of these differences solely to resolution issues, since resolution is not the only aspect in which the models differ.

The sentence has been modified: "To isolate the impact of dynamical downscaling on the projected changes"

L668-669: the authors state that the sea-level projections of IBI-CCS_raw and IBI-C'S'S_corr (note the typo) are very similar. However, this seems to be contradicted by Figures 15b-d, which show fairly large differences in the deep ocean. Since the differences in DSLC (Figure 15a) are smaller, apparently manometric SLC also differs substantially due to the bias corrections. Could the authors comment on this and explain why the bias corrections appear to affect the steric and manometric changes in the deep ocean in a substantial but compensating manner?

The sentence L668-669 is modified as follows: "The main spatial patterns of the differences between the projected changes of IBI-CCS_raw and IBI-CCS_corr are very similar for the two scenarios (Fig. 15), except in the Mediterranean Sea for the halosteric SL changes (Fig. 15d). Elsewhere, the impact of the corrections does not seem to be affected by the scenario." The manometric changes are not presented in Figure 15. The bias corrections appear to affect the thermosteric and halosteric changes in the deep ocean in a substantial but compensating manner at it is often the case according to Fox-Kemper et al., 2021: "Redistribution of water masses often involves anticorrelated thermosteric and halosteric changes especially in the Atlantic."

L689: significant differences or significant changes? These plots do not show differences between models do they?

Thank you for pointing this mistake out. Indeed, these plots show significant changes.

L700-702: could the authors point to any potential physical mechanism behind the decrease of the magnitude of interannual sea-level variability in the North Sea?

A sentence has been added: "The decrease in the magnitude of the sea level interannual variability in the North Sea is associated with a reduction of the CNRM-CM6-1-HR wind variability over the same region (not shown), consistently with Hermans et al., 2020a, who have shown that the sea level interannual variability is mainly driven by the atmospheric forcing variability in the region. This change in wind forcing does not appear to be a robust climate change signal under the SSP5-8.5 scenario in the CMIP6 GCMs ensemble, thus the change in sea level variability in the North Sea is probably a specific feature of CNRM-CM6-1-HR and thus IBI-CCS."

L749-750: it would be good to refer to Hermans et al. (2020) here once more, since they already argued the same thing, based on downscaling two GCMs differing in ocean grid resolution in the region

A sentence has been added L750: "Hermans et al., 2020b previously provided this argument based on the downscaling of two GCMs with different ocean grid resolution in the region."

L751-752: it would be good to add that to verify this, CNRM-CM6-1 would need to be downscaled as well

Done.

L776-777: Other GCMs may have other biases that may require different corrections. Different types of corrections, for example based on different reanalysis products, have not been tested here. Additionally, the model set-up in this manuscript requires more computational effort than time-slice methods such as used by Jin et al. (2021), which makes it difficult to obtain the large ensembles that are eventually required for comprehensive projections. Some more discussion of the caveats and the wider application of the methods in this study seems warranted.

A paragraph has been added L772: "The purpose of this paper was not to develop a new correction method, but rather to use a state-of-the-art method to develop our model configuration and simulations. However, this bias correction method assumes that biases are stationary, while several papers have shown the non-stationarity of biases (Maraun, 2012; Nahar et al., 2017; Hui et al., 2020). Another caveat is that 3D variables are independently corrected. However, the large amplitude of biases found in the GCM justifies the use of stationary seasonal mean bias corrections to address them. The bias correction method was chosen over the delta method (in which the mean climate change signal is added to the present-day time series) because it preserves the variability of the GCM. The internal variability of the regional model is thus driven by the GCM for the historical period and projections, allowing projected changes of the variability in the regional simulations to be investigated. More sophisticated bias correction methods are currently being developed (such as emergent constraints or multi-variable corrections). However, these methods are not yet mature enough and are potentially difficult to apply in the case of a 3D model."

The paragraph L784 has been revised considering your remarks: "The methodology used here to produce climate projections dealing with GCM biases is intended to be applied to a larger number of models in forthcoming studies. However, when considering a large number of models, the bias correction method may not be systematically applicable. Moreover, the model set-up requires more computational effort than time-slice methods such as that used by Jin et al. (2021), which makes it difficult to obtain the large ensembles that are eventually required for comprehensive projections."

Finally, the authors may consider adding another test, comparing a simulation with and without tides, for example in terms of seasonal biases and the simulation of mean sea-level change. This may be well outside the scope of the manuscript, but if relatively easily implemented in the IBI-CCS framework, it could give valuable insights into the limitations of models excluding tides in the context of projecting mean sea-level change.

Thank you for the interesting idea. This may be outside the scope of the manuscript but it might be interesting to run this simulation anyway. It could be relevant to have this simulation in the future study on the projections of extreme sea levels too.

**References**

Adloff, F., Somot, S., Sevault, F., Jordà, G., Aznar, R., Déqué, M., Herrmann, M., Marcos, M., Dubois, C., Padorno, E., Alvarez-Fanjul, E., and Gomis, D.: Mediterranean Sea response to climate change in an ensemble of twenty first century scenarios, Clim Dyn, 45, 2775–2802, https://doi.org/10.1007/s00382-015-2507-3, 2015.

[revised manuscript text omitted]

---

## Author Comment (AC2)

08.02.2022

Answer RC2:

The authors wish to thank the anonymous reviewer for his/her comments, questions and remarks that greatly helped us improving the quality of the paper. Please find below the point-by-point answers to your review in blue in the supplement to this comment.

Best regards,

The authors.

**Main concerns**

The title of the manuscript is misleading. This manuscript mainly focuses on assessing the impact on the modelled regional sea-level change of an increased horizontal resolution of the DD, a more complete representation of coastal processes, and applying a bias correction to the driving GCM. Using only a single GCM downscale is not sufficient to provide a reliable sea-level change estimation. I suggest changing the title of the manuscript to reflect the primary goal of the work.

We thank the reviewer for this comment highlighting that the title was not reflecting the aim of our work. The purpose of the paper is indeed not to provide a reliable sea-level change estimation not even to characterize uncertainties associated with the simulations. The study is meant to present a regional ocean model that will be used for analyzing the sensitivity of sea level changes, particularly extreme sea level changes, to methodological choices and representation of processes. In the current study, the configuration is presented along with its evaluation and the added value of the regional vs global model. It is from this perspective that we assess the impact of the dynamical downscaling and bias corrections on simulations of sea level changes. As the aim of the manuscript is to present the ocean regional model even more than to evaluate it, the title has been changed to "IBI-CCS: a regional high-resolution model to simulate sea level in Western Europe". The methodology presented in this paper could subsequently be applied to produce an ensemble of simulations using different CMIP6 global models as parent models to provide projections of sea level changes and related uncertainties.

Paragraph 2.1.2 Regional ocean model IBI-CCS: Tide in the regional configuration is one of the main processes driving SL change in coastal areas. A specific validation of the tides should be included in the manuscript. A reference to a peer-reviewed paper in which the tides have been validated is also enough. Moreover, the authors claim that "Tides are included in the model by calculating the astronomical tidal potential and the tidal harmonic forcing as …". Here the author should be more clear on the way they applied the tidal forcing in the regional model.

A sentence has been added in the manuscript: "Tides have been validated in section 3.1 of Maraldi et al., 2013 with a 1/36° configuration.". In Figure RC1, we provide the M2 tidal amplitude for FES2014 (a) and IBI-CCS_corr (b) and the difference FES2014 minus IBI-CCS_corr (c). In general, the regional model is close to the FES2004 solution, except north of the Irish Sea and in the German Bight. The figure has been added to the Supplementary Materials.

[Figure]

**Figure RC1: M2 tidal amplitude for FES2014 (a) and IBI-CCS_corr (1993-2014) (b) and the difference FES2014 minus IBI-CCS_corr (c) for the coastal points of the zone.**

Also, the paragraph explaining how the tidal forcing is applied in the regional model has been revised: "Tides are included in the model by calculating the astronomical tidal potential and the tidal harmonic forcing. SSH and barotropic velocities tidal components are added through the open boundaries as the sum of 11 components provided by FES2004 (Lyard et al., 2006) and TPXO7.1 (Egbert and Erofeeva, 2002): diurnal components (K1, O1, P1 and Q1), semi-diurnal constituents (M2, S2, N2 and K2), long-period tides (Mf and Mm) and a nonlinear component M4."

Paragraph 2.3.2. In line 308, the authors state that "The GCM GMTSLR term stored in the variable "zostoga" is thus added a posteriori to the RCM modelled SL". My deep concern is how the authors used this variable in the final SL computation (Figure 9). The global model used in this study is affected by strong temperature (and salinity) drift due to its relatively short spin-up (250 yrs). In particular, the temperature drift affects the local thermosteric component of the SL, and so the global mean thermosteric component (zostoga). Maybe I am wrong, but it seems that the authors used the original zostoga variable provided by the global simulation without any correction. I suggest the authors to indicate in the manuscript how they treated zostoga before using it in the final SL computation.

Thank you for pointing this omission. The original variable zostoga has not been used for the reasons you mentioned. The variable is indeed corrected from the drift based on the same method as the corrections applied to the open boundary conditions. The drift is estimated by a linear fit of the full time series of the pre-industrial control simulation (Gupta et al., 2013). Then, the linear fit is subtracted to the corresponding historical simulation and projections at each time step. This method has been used in recent studies for the variable "zostoga" for example in Hermans et al., 2021 and Fox-Kemper et al., 2021 (Supp Mat Chap 9, section 9.SM.4.2). In Figure RC2, we provide the original and corrected variable "zostoga" (monthly outputs) for the two scenarios:

[Figure]

**Figure RC2: Original and drift corrected global mean thermosteric sea level over the 1950-2100 period for the two scenarios SSP5-8.5 and SSP1-2.6.**

The information has been added in:

- section 2.2.1 "IBI-CCS_corr simulation": "The drift is also removed from the global mean thermosteric sea level (variable "zostoga") of section 2.3.2 using the same method.".
- section 2.3.2 "Thermal expansion": "The GCM GMTSL term stored in the variable "zostoga" is thus added a posteriori to the RCM modeled SL after having removed the drift (section 2.2.1)."

Line 172: I suspect that mixing due to internal tides is overestimated. So, I suggest to provide more details about the de Lavergne scheme and a more robust justification about its use in the regional model.

Despite the regional ocean model explicitly resolves tides, the entire spectrum of internal waves is not generated especially at a 1/12° resolution. Actually, at this resolution, only the most energetic modes (mode 1 and 2) of the internal tides are resolved. In the IBI zone, the mode 1 seems to be dominant internal tide mode (Vic et al., 2019) so in our case we might generate a large part of the locally generated internal tide spectrum. However, the model doesn't account for low modes propagating into the IBI region at its boundaries, nor does it have the required

physics to dissipate explicitly resolved internal tides correctly. As stated in Melet et al., 2022, "even ocean general circulation models with explicit tides typically do not resolve the generation of high-mode internal tides, scattering of low-mode energy into higher modes, and various processes leading to internal-tide energy dissipation, so parameterizations are still required to get realistic internal tides and dissipation (e.g Arbic et al., 2010; Ansong et al., 2017)." For these reasons, we have applied the de Lavergne et al., 2020 parametrization which is at the state-of-the-art in terms of representation of internal tide induced mixing processes (hence allowing for a molecular background diapycnal mixing).

Line 249: Sea Surface Height tuning in the Mediterranean Sea. The authors claim that **"GLORYS2V4 has a mean SSH bias of approximately -0.1 m in the Mediterranean Sea in comparison to the Mean Dynamic Topography observations from CNES- CLS-18"**. It would be good to show, at least in the supplementary material, the horizontal map showing the differences between GLORYS2V4 and CNES- CLS-18 over the entire domain. The -0.1 value used as a correction seems to result from a tuning exercise. The authors should provide more details on the applied correction if this is the case. Also, looking at Figure 3, it appears that in all simulations (including IBI-CCS_corr) there is a bias on the eastern boundary in the Mediterranean Sea. Do the authors have a valid justification for the bias in IBI-CCS_corr?

In Figure RC3, we provide the map showing the bias (c) between GLORYS2V4 (1993-2014) (b) and CNES-CLS-18 (1993-2012) (a) over the entire domain. The bias in the Mediterranean Sea is approximately of 10cm. The figure has been added to the supplementary materials as well. The bias in the Mediterranean Sea of approximately of +10 cm is due to the assimilation of two different sea level anomaly databases in GLORYS2V4: a global one for the Atlantic part and a Mediterranean one. These two databases were not aligned with each other.

[Figure]

**Figure RC3: Sea Surface Height bias (c) between GLORYS2V4 (1993-2014) (b) and CNES-CLS-18 (1993-2012) (a).**

As explained L260, the SSH corrective value of +0.1 m is applied to the east Mediterranean boundary at each time step and boundary grid point. A sentence has been added to detail how the correction is applied: "The mass correction is added to the local T/S values."

The +0.1m setting of the SSH is applied in both IBI-CCS_corr and in IBI-ERAi simulations so the biases observed at the eastern boundary in Figure 3 are not related to this setting. The biases of Figure 3 are found in the buffer zone where the bathymetries of CNRM-CM6-1-HR (1/4°) and of the regional model (1/12°) are merged. However, for the IBI-ERAi simulation, the forcings at the open boundaries are GLORYS2V4 (and not CNRM-CM6-1-HR as in IBI-CCS simulations), so there is a small inconsistency in the buffer zone between the forcings and bathymetry. That is why biases are observed for integrated variables like thermosteric or halosteric sea level.

As a general comment, the manuscript needs revision for language and grammar.

The manuscript has been revised for language and grammar.

**Minor issues**

Line 13: Please, include the name of the model "(Iberian-Biscay-Ireland Climate Change Scenarios)"

Done.

Line 63: Please, provide the physical definition for "dynamic sea level".

The physical definition has been added above L41: "At regional scales, spatial variations of SL changes are mainly due to changes in dynamic sea level (DSL) i.e. changes in ocean circulations and the associated ocean heat, salt and mass redistribution within the ocean."

Line 75: The authors claim that "The DD method can be used to overcome this problem by applying corrections to the GCM outputs before using them as forcing when performing a DD". Actually, the bias in GCM simulations can be strongly reduced using bias correction. So, I do not agree in 'the DD methods'. May be this sentence need to be revised or deleted.

The sentence has been modified as follows: "To overcome this problem, bias corrections can be applied to the GCM outputs before using them as forcing when performing a dynamical downscaling"

Line 114: Paragraph 2.1.1. It would be good to add a specific subparagraph in which is indicated how the SSH is modelled in both global models.

The subparagraph "2.3.3 Total SL in global and regional simulations" has been modified and explains how the SSH is modelled in both global and regional models.

Line 115: I did not find any specific paper in the literature dedicated to the validation of CNRM-CM6-1-HR. Am I wrong? In case you could not provide any reference to the validation of CNRM-CM6-1-HR it would be necessary to justify the use of this model simulation as driver for the DD.

Indeed only its lower resolution version (CNRM-CM6-1) has been extensively validated in Voldoire et al., 2019. The CNRM-CM6-1-HR model has been derived from its lower resolution counterpart by only increasing the resolution. This high-resolution version provides results very close to CNRM-CM6-1 which has not motivated a dedicated paper. However, in Saint-Martin et al., 2021, there is a brief description of this model and the supplement information material provides some figures comparing the biases of both model versions (https://agupubs.onlinelibrary.wiley.com/action/downloadSupplement?doi=10.1029%2F2020MS002190&file=2020MS002190-sup-0002-Supporting+Information+SI-S01.pdf). An explanation has been provided L115 to justify the use of this GCM: "The GCM CNRM-CM6-1-HR was chosen for its eddy-permitting high-resolution over the ocean which allows a more realistic regional circulation (section 3.1.2). Moreover, the 0.5° resolution over the atmosphere is also interesting to obtain a less smooth atmospheric forcing (winds) which is very important for the modeling of extreme SLs."

Line 122: The following sentence is not clear to me: "A polynomial representation of the equation of state (TEOS-10, Roquet et al., 2015) is used but the temperature and salinity outputs are converted into the in-situ temperature and practical salinity needed by the RCM. ". Please, rewrite this sentence.

The sentence L122 has been modified: "Seawater thermodynamics uses a polynomial approximation of TEOS-10 (Roquet et al., 2015), therefore the prognostic variables are the absolute salinity and conservative temperature. As the RCM does not use the same approximation for the equation of state (section 2.1.2), the GCM outputs are converted to in-situ temperature and practical salinity to be used in the RCM."

The sentence L171 has been modified: "Seawater thermodynamics uses a polynomial approximation of EOS-80 (Fofonoff and Millard Jr, 1983). Therefore, the RCM requires in-situ temperature and practical salinity from the GCM."

Line 128: please, insert citation for OASIS-MCT

The citation « Craig et al., 2017 » has been added.

Line 129 please, insert a citation for ARPEGE-Climat 6.3

The citation « Roehrig et al., 2020 » has been added.

Line 130: I would suggest to indicate the exact number of simulations used.

Done.

Line 138: In this paragraph should be indicated the model resolution of the regional model

The title of the section has been changed in "Regional ocean model IBI-CCS at 1/12° resolution".

Line 151: I don't think it is relevant to indicate the 1/36° version in the manuscript. It is not used for validation. So, I suggest to remove it.

The 1/36° version is indeed not used for validation. This indication gives an information about the knowledge of the area in the CMEMS framework.

Line 152: Please, provide the explicit link in the references.

Done

Line 230: Please, provide the link in the references

Done.

Line 240: The paragraph is not enough clear. Please, rewrite this paragraph.

Done.

**References**

Ansong, J., Arbic, B., Alford, M., Buijsman, M., Shriver, J., Zhao, Z., Richman, J., Simmons, H., Timko, P., Wallcraft, A., and Zamudio, L.: Semidiurnal internal tide energy fluxes and their variability in a Global Ocean Model and moored observations, Journal of Geophysical Research: Oceans, 122, https://doi.org/10.1002/2016JC012184, 2017.

Arbic, B. K., Wallcraft, A. J., and Metzger, E. J.: Concurrent simulation of the eddying general circulation and tides in a global ocean model, Ocean Modelling, 32, 175–187, https://doi.org/10.1016/j.ocemod.2010.01.007, 2010.

Craig, A., Valcke, S., and Coquart, L.: Development and performance of a new version of the OASIS coupler, OASIS3-MCT_3.0, 10, 3297–3308, https://doi.org/10.5194/gmd-10-3297-2017, 2017.

Egbert, G. D. and Erofeeva, S. Y.: Efficient Inverse Modeling of Barotropic Ocean Tides, 19, 183–204, https://doi.org/10.1175/1520-0426(2002)019<0183:EIMOBO>2.0.CO;2, 2002.

Fofonoff, N. P. and Millard Jr, R. C.: Algorithms for the computation of fundamental properties of seawater., https://doi.org/10.25607/OBP-1450, 1983.

Fox-Kemper, B., Hewitt, H.T., Xiao, C., Aðalgeirsdóttir, G., Drijfhout, S.S., Edwards, T.L., Golledge, N.R., Hemer, M., Kopp, R.E., Krinner, G., Mix, A., Notz, D., Nowicki, S., Nurhati, I.S., Ruiz, L., Sallée, J.-B., Slangen, A.B.A., and Yu, Y.: Ocean, Cryosphere and Sea Level Change Supplementary Material. In Climate Change 2021: The Physical Science Basis. Contribution of Working Group I to the Sixth Assessment Report of the Intergovernmental Panel on Climate Change [MassonDelmotte, V., Zhai, P., Pirani, A., Connors, S.L., Péan, C., Berger, S., Caud, N., Chen, Y., Goldfarb, L., Gomis, M.I., Huang, M., Leitzell, K., Lonnoy, E., Matthews, J.B.R.,

Maycock, T.K., Waterfield, T., Yelekçi, O., Yu, R., and Zhou, B. (eds.)]. Cambridge University Press. In Press. 2021

Gupta, A. S., Jourdain, N. C., Brown, J. N., and Monselesan, D.: Climate Drift in the CMIP5 Models, 26, 8597–8615, https://doi.org/10.1175/JCLI-D-12-00521.1, 2013.

Hermans, T. H. J., Gregory, J. M., Palmer, M. D., Ringer, M. A., Katsman, C. A., and Slangen, A. B. A.: Projecting Global Mean Sea-Level Change Using CMIP6 Models, 48, e2020GL092064, https://doi.org/10.1029/2020GL092064, 2021.

Lyard, F., Lefevre, F., Letellier, T., and Francis, O.: Modelling the global ocean tides: modern insights from FES2004, Ocean Dynamics, 56, 394–415, https://doi.org/10.1007/s10236-006-0086-x, 2006.

Maraldi, C., Chanut, J., Levier, B., Ayoub, N., De Mey, P., Reffray, G., Lyard, F., Cailleau, S., Drévillon, M., Fanjul, E. A., Sotillo, M. G., and Marsaleix, P.: NEMO on the shelf: assessment of the Iberia-Biscay-Ireland configuration, All Depths/Operational Oceanography/All Geographic Regions/Temperature, Salinity and Density Fields, https://doi.org/10.5194/osd-10-83-2013, 2013.

Melet, A. V., Hallberg, R., and Marshall, D. P.: Chapter 2 - The role of ocean mixing in the climate system, in: Ocean Mixing, edited by: Meredith, M. and Naveira Garabato, A., Elsevier, 5–34, https://doi.org/10.1016/B978-0-12-821512-8.00009-8, 2022.

Roehrig, R., Beau, I., Saint-Martin, D., Alias, A., Decharme, B., Guérémy, J.-F., Voldoire, A., Abdel-Lathif, A. Y., Bazile, E., Belamari, S., Blein, S., Bouniol, D., Bouteloup, Y., Cattiaux, J., Chauvin, F., Chevallier, M., Colin, J., Douville, H., Marquet, P., Michou, M., Nabat, P., Oudar, T., Peyrillé, P., Piriou, J.-M., Salas y Mélia, D., Séférian, R., and Sénési, S.: The CNRM Global Atmosphere Model ARPEGE-Climat 6.3: Description and Evaluation, 12, e2020MS002075, https://doi.org/10.1029/2020MS002075, 2020.

Roquet, F., Madec, G., Brodeau, L., and Nycander, J.: Defining a Simplified Yet "Realistic" Equation of State for Seawater, 45, 2564–2579, https://doi.org/10.1175/JPO-D-15-0080.1, 2015.

Saint-Martin, D., Geoffroy, O., Voldoire, A., Cattiaux, J., Brient, F., Chauvin, F., Chevallier, M., Colin, J., Decharme, B., Delire, C., Douville, H., Guérémy, J.-F. -f, Joetzjer, E., Ribes, A., Roehrig, R., Terray, L., and Valcke, S.: Tracking Changes in Climate Sensitivity in CNRM Climate Models, 13, https://doi.org/10.1029/2020ms002190, 2021.

Vic, C., Naveira Garabato, A. C., Green, J. A. M., Waterhouse, A. F., Zhao, Z., Melet, A., de Lavergne, C., Buijsman, M. C., and Stephenson, G. R.: Deep-ocean mixing driven by small-scale internal tides, Nat Commun, 10, 2099, https://doi.org/10.1038/s41467-019-10149-5, 2019.

Voldoire, A., Saint-Martin, D., Sénési, S., Decharme, B., Alias, A., Chevallier, M., Colin, J., Guérémy, J.-F., Michou, M., Moine, M.-P., Nabat, P., Roehrig, R., Mélia, D. S. y, Séférian, R., Valcke, S., Beau, I., Belamari, S., Berthet, S., Cassou, C., Cattiaux, J., Deshayes, J., Douville, H., Ethé, C., Franchistéguy, L., Geoffroy, O., Lévy, C., Madec, G., Meurdesoif, Y., Msadek, R., Ribes, A., Sanchez-Gomez, E., Terray, L., and Waldman, R.: Evaluation of CMIP6 DECK Experiments With CNRM-CM6-1, 11, 2177–2213, https://doi.org/10.1029/2019MS001683, 2019.